Methods

# A lyophilized open-source RT-LAMP assay for molecular diagnostics in resource-limited settings

Martin Matl[1,2,*], Max J Kellner[1,2,3,4,*], Felix Ansah[5,*], Irina Grishkovskaya[1], Dominik Handler[2], Robert Heinen[1,2], Benedikt Bauer[1], Luis Menéndez-Arias[6], Thomas O Auer[7,8], Lucia L Prieto-Godino[8,9], Josef M Penninger[2,4,10], Vienna Covid-19 Detection Initiative (VCDI), Gordon A Awandare[5], Julius Brennecke[2], Andrea Pauli[1]

**A critical bottleneck for equitable access to population-scale molecular diagnostics is the limited availability of rapid, inexpensive point-of-care tests, especially in low- and middle-income countries. Here, we developed an open-source reverse transcription loop–mediated isothermal amplification (RT-LAMP) molecular assay for pathogen detection. It is based on non-proprietary enzymes, namely, HIV-1 reverse transcriptase, _Bst_ LF DNA polymerase, and UDG BMTU thermolabile uracil-DNA glycosylase. Formulated as liquid or lyophilized reaction mixtures, these reagents enable sensitive colorimetric detection of respiratory samples without the need for prior nucleic acid isolation. We evaluated our lyophilized RT-LAMP assay on clinical samples with suspected COVID-19 infection, demonstrating high sensitivity and 100% specificity compared with the gold-standard RT–qPCR. Reaction performance was unaffected by prolonged storage of lyophilized reagents at ambient or elevated temperatures. As a proof of concept, we evaluated the robustness and ease of use of lyophilized RT-LAMP reaction mixes through independent laboratory testing of COVID-19 samples in Ghana. Overall, our open-source RT-LAMP assay provides a flexible and scalable point-of-care test that can be adapted for rapid detection of various pathogens in resource-limited settings.**

## Introduction

In vitro nucleic acid testing is an essential tool for containing outbreaks of infectious viral diseases (Perkins et al, 2017; Peeling et al, 2022). The COVID-19 pandemic has highlighted the unpreparedness of countries to rapidly establish testing programs, in part because of inadequate laboratory infrastructure to perform molecular testing (Mfuh et al, 2023). In addition, molecular diagnostic testing is commonly linked to the use of proprietary reagents, creating a dependency on commercial solutions. Although optimized to ensure robust performance, population-scale testing with commercial kits can be prohibitively expensive, especially for low- and middle-income countries that suffer from underinvestment in research and development (Petti et al, 2006). These countries also often have limited access to international supply chains for reagent distribution (Nkengasong, 2020; WHO, 2023). To address these challenges, several programs have been established to ensure equitable access to health products for all countries, such as the World Health Organization's (WHO) COVID-19 Technology Access Pool (Venkatesan, 2023). However, key challenges remain to ensure the development and self-sufficient implementation of molecular testing programs for resource-limited settings (Jani & Peter, 2022).

Reverse transcription loop–mediated isothermal amplification (RT-LAMP) is a powerful molecular diagnostic tool with the potential to overcome many of the challenges of in vitro nucleic acid diagnostics (Notomi et al, 2000; Feddema et al, 2024). As an isothermal DNA amplification technique, RT-LAMP enables rapid and sensitive detection of RNA and DNA templates without the need for sophisticated laboratory infrastructure (Notomi et al, 2000; Kellner et al, 2022). Unlike conventional PCR, RT-LAMP assays are also compatible with colorimetric readouts (Goto et al, 2009; Tanner et al, 2015; Scott et al, 2020). Detection by visual color change can be particularly advantageous in resource-limited settings, as the binary reaction outcome (i.e., pathogen positive or negative) can be interpreted even by laypersons without specialized equipment.

[1]Research Institute of Molecular Pathology (IMP), Vienna BioCenter (VBC), Vienna, Austria [2]Institute of Molecular Biotechnology of the Austrian Academy of Sciences (IMBA), Vienna BioCenter (VBC), Vienna, Austria [3]Vienna BioCenter PhD Program, Doctoral School of the University at Vienna and Medical University of Vienna, Vienna, Austria [4]Department of Laboratory Medicine, Medical University of Vienna, Vienna, Austria [5]West African Center for Cell Biology of Infectious Pathogens (WACCBIP), College of Basic and Applied Sciences, University of Ghana, Legon, Ghana [6]Centro de Biología Molecular 'Severo Ochoa' (Consejo Superior de Investigaciones Científicas and Universidad Autónoma de Madrid), Madrid, Spain [7]Department of Biology, University of Fribourg, Fribourg, Switzerland [8]TReND in Africa, Brighton, UK [9]The Francis Crick Research Institute, London, UK [10]Helmholtz Centre for Infection Research, Braunschweig, Germany

Correspondence: gawandare@ug.edu.gh; julius.brennecke@imba.oeaw.ac.at; andrea.pauli@imp.ac.at
Max J Kellner's present address is Helmholtz Centre for Infection Research, Braunschweig, Germany
*Martin Matl, Max J Kellner, and Felix Ansah contributed equally to this work

Most commercially available RT-LAMP products to date contain proprietary, engineered enzymes and reaction components optimized to achieve high sensitivity and specificity (Ong et al., 2015; McMahon, 2021; Sharma et al., 2024). However, researchers and medical professionals in resource-limited settings often cannot afford these commercial solutions (2.9–6$/reaction) or are located in countries without local distribution partners (Nkengasong, 2020; Ondoa et al, 2020). To overcome this, we and others have provided open-source protocols and reagents for low-cost RT-LAMP assays (Tomita et al, 2008; Bhadra et al, 2018; Alekseenko et al, 2021; Kellner et al, 2022). Nevertheless, easily deployable complete formulations, independent of cold chain storage of enzymes, reagents, and oligonucleotide primers, are urgently needed for point-of-care diagnostics and for monitoring applications in resource-limited settings (Jani & Peter, 2022). Here, we present a sensitive and robust, fully open-source RT-LAMP system that meets all these criteria and allows colorimetric RT-LAMP from lyophilized reagents. Detailed protocols for preparing RT-LAMP enzymes and running assays are also provided on our website rtlamp.org.

## Results

### Open-source RT-LAMP enables sensitive and specific RNA detection

The performance of RT-LAMP assays is strongly dependent on the enzyme mix used, especially the type of reverse transcriptase (Kellner et al, 2022). To establish a low-cost, open-source RT-LAMP protocol for RNA detection, we therefore sought to determine which nonproprietary enzymes provide optimal reaction performance, using SARS-CoV-2 RNA as the target and validated RT-LAMP primers for benchmarking (Rabe & Cepko, 2020; Kellner et al, 2022). We first compared several variants of the commonly used thermophilic DNA polymerase from *Geobacillus stearothermophilus*, commercially available *Bst* LF (New England Biolabs), engineered *Bst* 2.0 DNA polymerase (NEB), and the improved DNA-binding and salt-tolerant *Bst* 3.0 (NEB) against our in-house–purified WT *Bst* LF (Fig S1A, C, G). Using a synthetic SARS-CoV-2 genomic RNA dilution series as input, we observed comparable reaction specificity and sensitivity among all tested *Bst* polymerase variants at sample concentrations as low as 25–50 RNA copies/μl (Fig 1A). Overall, in-house–produced *Bst* LF showed a performance similar to engineered *Bst* variants on synthetic SARS-CoV-2 genomic RNA (Figs 1A and S2A).

Next, we determined which reverse transcriptase (RT) provides optimal reaction performance in combination with our in-house–produced *Bst* LF polymerase. To this end, we compared open-source, in-house–produced HIV-1 RT (Fig S1B–D and G) with more commonly used RT enzymes, Moloney murine leukemia virus RT (M-MuLV RT, NEB), avian myeloblastosis virus RT (AMV RT, NEB), and engineered WarmStart RTx (NEB) (Fig 1B). Of all the reverse transcriptases, WarmStart RTx provided the shortest onset of amplification (Fig S2B) and showed the highest overall sensitivity (Fig 1B). Notably, our in-house–purified HIV-1 RT showed a

performance on par with WarmStart RTx in the tested conditions, both detecting 4/4 replicates at 50 RNA copies/μl (Figs 1B and S2B). We conclude that HIV-1 RT is the enzyme of choice for open-source RT-LAMP assays.

Nucleic acid contamination is a serious risk associated with LAMP assays that can compromise test results and shut down entire diagnostic operations (Robinson-McCarthy et al, 2021). Uracil-DNA glycosylase (UDG) enzymes are known to effectively prevent contamination in conjunction with dUTP in RT-LAMP assays (Jaeger et al, 2000; Hsieh et al, 2014; Kellner et al, 2022). To reduce the risk of cross-contamination, we therefore produced the thermolabile uracil-DNA glycosylase from the psychrotrophic marine bacterium BMTU 3346 (hereafter referred to as BMTU UDG) (Fig S1E–G). Reaction performance in comparison with commercial Antarctic Thermolabile UDG (NEB) was tested in forced contamination experiments, where diluted amounts of dUTP-containing SARS-CoV-2 LAMP amplicons were added to RT-LAMP reactions containing synthetic SARS-CoV-2 RNA. As expected, the absence of UDG resulted in indistinguishable amplification between positive and negative samples regardless of the amount of contaminating amplicon added (Fig 1C). In contrast, in-house–produced or commercial thermolabile UDG added directly to the RT-LAMP reactions effectively prevented amplification in contaminated reactions in an amplicon concentration–dependent manner without substantially reducing reaction sensitivity (Figs 1C and S2C). Importantly, we confirmed that the prevention of contamination was based on the incorporation of dUTP, as dTTP-containing amplicons were amplified indiscriminately from the true RNA template regardless of the presence or absence of UDG (Fig 1D). Moreover, preincubation of in-house UDG above 50°C abolished its anticontamination function, confirming the thermolabile nature of BMTU UDG (Fig 1D).

To benchmark our complete open-source RT-LAMP reaction mix containing only in-house–produced enzymes (HIV-1 RT, *Bst* LF, and BMTU UDG) with the gold-standard commercial RT-LAMP reagents from New England Biolabs, we used a dilution series of synthetic SARS-CoV-2 RNA. To establish a robust detection limit, RT-LAMP assays were performed in 20 replicates. The open-source RT-LAMP reagents resulted in a 100% (20/20) detection rate at 50 RNA copies/μl and an 85% (17/20) detection rate at 25 RNA copies/μl, which is comparable to the sensitivity measured for the commercial RT-LAMP reagents (Fig 1E).

The sensitivity of the RT-LAMP assay can be further enhanced by a simple, 15-min magnetic bead enrichment step that effectively concentrates the input nucleic acid material (Fig S2D) We adapted this "bead-LAMP" protocol to make it compatible with our open-source enzymes by (i) increasing the amount of polymerase present in the reaction mix, and (ii) using in-house–manufactured nucleic acid capture beads (Oberacker et al, 2019). Bead-LAMP with open-source reagents on dilutions of synthetic SARS-CoV-2 RNA (Twist Bioscience) allowed reliable detection of 12.5 copies/μl (4/4 detected) and 3.125 copies/μl (3/4 detected) of the original sample concentration (Fig S2E), demonstrating that the sensitivity of open-source RT-LAMP can be further increased with bead-LAMP. The specificity of our open-source RT-LAMP reaction mix was evaluated by adding either SARS-CoV-2 RNA, influenza A RNA, or water to purified RNA

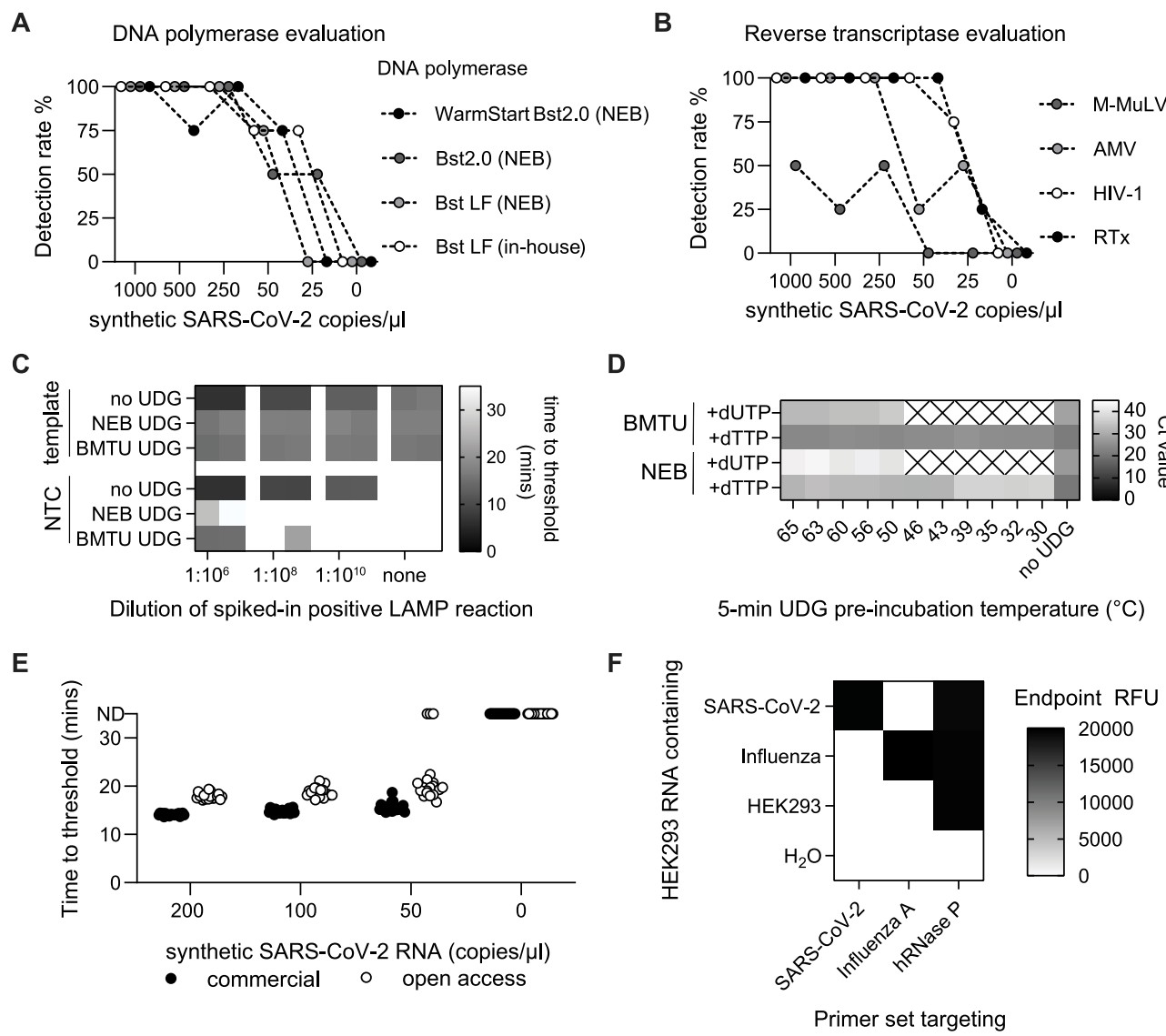

**Figure 1. Assessing the performance of open-source RT-LAMP enzymes.**
**(A)** Comparison of DNA polymerase enzymes in RT-LAMP reactions. Detection rates of RT-LAMP reactions containing different DNA polymerases in addition to WarmStart RTx reverse transcriptase (NEB) and various concentrations of synthetic SARS-CoV-2 RNA. Results were obtained from four replicates per condition. **(B)** Comparison of reverse transcriptase enzymes in RT-LAMP reactions. Analogous to (A) but for different reverse transcriptase enzymes in combination with in-house *Bst* LF DNA polymerase. M-MuLV, Moloney murine leukemia virus reverse transcriptase; AMV, avian myeloblastosis virus reverse transcriptase; HIV-1, human immunodeficiency virus 1 reverse transcriptase. **(C)** Cross-contamination prevention in RT-LAMP via uracil-DNA glycosylase (UDG) enzymes. Diluted amounts of contaminating amplicons were added to RT-LAMP reactions containing different uracil-DNA glycosylase (UDG) enzymes and synthetic SARS-CoV-2 template. No UDG enzyme and nontemplate control reactions were included. Shown are time-to-threshold values from real-time fluorescence RT-LAMP reactions performed in duplicates. In-house BMTU UDG enzyme was tested against Antarctic Thermolabile UDG (NEB). **(D)** Thermostability test of BMTU UDG and commercial UDG enzymes. In-house BMTU UDG and commercial Antarctic Thermolabile UDG (NEB) were preincubated at different temperatures for 5 min. UDG enzyme was then added to qPCRs targeting either dTTP-containing DNA template or dUTP-containing DNA template. Cycle-to-threshold values (Ct) from duplicate reactions are shown for each condition. Crossed box = not determined. **(E)** Limit of detection of open-access and commercial RT-LAMP reactions. Reactions prepared from in-house *Bst* LF, HIV-1 RT, and BMTU UDG enzymes were compared with commercial reactions using 2X WarmStart LAMP Kit (NEB) containing engineered proprietary enzymes. Synthetic SARS-CoV-2 RNA at defined copy numbers was used as a template, and 20 replicates were performed per condition. Time-to-threshold values from real-time fluorescence RT-LAMP reactions are shown. **(F)** Specificity test of RT-LAMP reactions detecting different pathogens. RT-LAMP reactions were assembled with *Bst* LF and HIV-1 RT. Primers targeting SARS-CoV-2, Influenza A, or hRNase P were tested against SARS-CoV-2 RNA, Influenza A RNA (each individually spiked into HEK293-extracted RNA), HEK293-extracted RNA alone, and nuclease-free water as a no-template control. Reactions were performed in four replicates, and end-point relative fluorescent units are displayed.
Source data are available for this figure.

extracted from HEK293 cells. RT-LAMP assays were tested for SARS-CoV-2 (Rabe & Cepko, 2020), influenza A (Takayama et al, 2019), or RNase P targeting the human RNase P transcript as an internal positive control (Broughton et al, 2020). Amplification was specific for each added RNA, and no cross-reactivity for either viral pathogen was observed (Fig 1F).

Taken together, we have established an open-source RT-LAMP reagent mix that enables rapid, sensitive, and specific colorimetric RNA detection and that performs on par with commercially available RT-LAMP reagents on contrived RNA samples. The produced enzymes were quality-controlled using activity assays as a simple quality control step, and were titrated to achieve optimum performance under our assay conditions (Fig S2F–H). To enable low-cost and open distribution of the open-source enzymes, we have deposited *E. coli* expression vectors for His-tagged versions of these three WT enzymes with Addgene (www.addgene.org) and provide standard protocols for medium-scale protein expression and purification in this article (see the Materials and Methods section) and on our website (rtlamp.org).

### A sample inactivation procedure compatible with nucleic acid detection

In most approved nucleic acid diagnostic tests, RNA extraction is a necessary preanalytical step. However, RNA extraction is a bottleneck in the supply chain, adds a significant cost per sample, and is time- and labor-intensive. Protocols that bypass the RNA extraction step have been reported previously (Myhrvold et al, 2018; Fomsgaard & Rosenstierne, 2020; Rabe & Cepko, 2020; Ulloa et al, 2020; Sandri et al, 2021). A 5-min heat inactivation step using a commercially available QuickExtract DNA Extraction Solution (Lucigen) has been shown to result in sensitive SARS-CoV-2 nucleic acid detection, including for RT–qPCR and RT-LAMP (Joung et al, 2020 *Preprint*; Kellner et al, 2022). We therefore aimed to develop an optimized, open-source, rapid sample inactivation protocol using readily available reagents to replace the proprietary QuickExtract solution.

The addition of the reducing agent tris(2-carboxyethyl)phosphine (TCEP) has been reported to be beneficial for preserving sample integrity after heat inactivation, especially of released naked RNA (Myhrvold et al, 2018; Rabe & Cepko, 2020). Therefore, we compared various TCEP-containing inactivation solutions with the commercial QuickExtract solution to develop a scalable, cost-effective alternative. Patient-derived SARS-CoV-2–positive nasopharyngeal swab and gargle samples were mixed with inactivation solutions and heat-inactivated at 95°C for 5 min (Fig 2A). The inactivated samples were then stored at different temperatures (RT, 4°C, and −20°C) and durations (0, 4, and 16 h) to simulate potential sample storage conditions. The integrity of viral RNA within the samples was assessed by direct-input RT–qPCR (Joung et al, 2020 *Preprint*; Kellner et al, 2021 *Preprint*). Although heat-inactivated swab samples collected in viral transport medium (VTM) without the addition of any inactivation buffer showed temperature-dependent degradation of viral RNA (Fig 2B), gargle samples in saline or HBSS proved more stable (Fig 2C), suggesting an intrinsically higher RNA stability in media lacking fetal bovine serum (FBS). Although effective in preventing RNA degradation, the addition of TCEP resulted in sample heterogeneity and turbidity after the 5-min heat inactivation step, presumably from precipitated protein with a likely negative impact on RT-LAMP colorimetric detection (Fig 2D). Supplementation of the TCEP inactivation reagent with betaine and proteinase K overcame this limitation, allowing for sensitive direct-input RT–qPCR and HNB colorimetric

RT-LAMP without affecting performance (Fig 2D–F). Overall, this demonstrates that a TCEP/betaine/proteinase K inactivation solution enables rapid sample inactivation under conditions that preserve RNA integrity and maintain compatibility with downstream direct-input RT–qPCR and RT-LAMP.

### Lyophilization of open-source RT-LAMP enzymes and reaction mixes

Having established a direct-input RT-LAMP assay using only open-source reagents, we next sought to develop a lyophilized mixture to enable global distribution independent of cold chain logistics. We established two separate lyophilization protocols using glycerol-free open-source RT-LAMP enzymes and D-trehalose as a cryoprotectant (Carter et al, 2017): a lyo-RT-LAMP enzyme mix containing only open-source enzymes ("enzyme mix"), which upon reconstitution provides a 20x concentrated glycerol enzyme mixture for immediate use or storage at −20°C. We also developed a lyo-RT-LAMP reagent mix containing enzymes, dNTPs, and primers for immediate use ("reagent mix") (Fig 3A) (see the Materials and Methods section and rtlamp.org for details).

To measure the performance of lyophilized enzymes after long-term storage, we reconstituted the freeze-dried lyo-RT-LAMP mixtures stored for 10 and 30 d at different temperatures (−22°C, 4°C, RT) and compared the reaction sensitivity with a freshly prepared enzyme mixture using a contrived, rapidly inactivated dilution series of a SARS-CoV-2–positive sample. We observed little to no measurable loss of sensitivity and no difference in reaction onset (time to threshold) in RT-LAMP reactions prepared with reconstituted enzymes compared with fresh reagents even after 30 d of storage (Figs 3B and S3A). Similar results were obtained for the lyophilized "reagent mix", with no change in reaction performance after 10 and 40 d of storage at different temperatures for contrived crude SARS-CoV-2 or synthetic SARS-CoV-2 RNA dilutions (Figs 3C and D and S3B and C). In addition, a cross-contamination test showed that BMTU UDG did not lose activity during lyophilization and was able to protect against cross-contamination in this temperature-stable formulation (Fig 3E).

To demonstrate the feasibility of providing open-source reagent mixes for home and remote point-of-care settings, we prepared individually lyophilized reagent mixes in PCR strips (1 reaction/tube) (Fig 3F). Such a lyophilized PCR strip with reagent mix was reconstituted with reconstitution buffer and tested in parallel with an RT-LAMP mix prepared from cold-stored enzymes (stored in glycerol-containing storage buffer at −20°C). The strips were tested on the same sample, demonstrating robust colorimetric reaction performance in lyophilized and cold-stored RT-LAMP reagent mixes (Fig 3F).

Finally, we aimed to demonstrate the utility of lyo-RT-LAMP reagent mixes for the detection of viral pathogens beyond SARS-CoV-2. In a proof-of-concept experiment, we assembled a multi-pathogen respiratory virus RT-LAMP test for rapid testing at the point of care, targeting SARS-CoV-2, human coronavirus NL63 (Amsterdam I), influenza A virus (H1N1 A/WSN/1933), respiratory syncytial virus (RSV A2), and human RNase P in a single eight-well PCR strip containing lyo-RT-LAMP reagent mixes with primer sets specific to these viruses. The multi-pathogen–lyophilized RT-LAMP

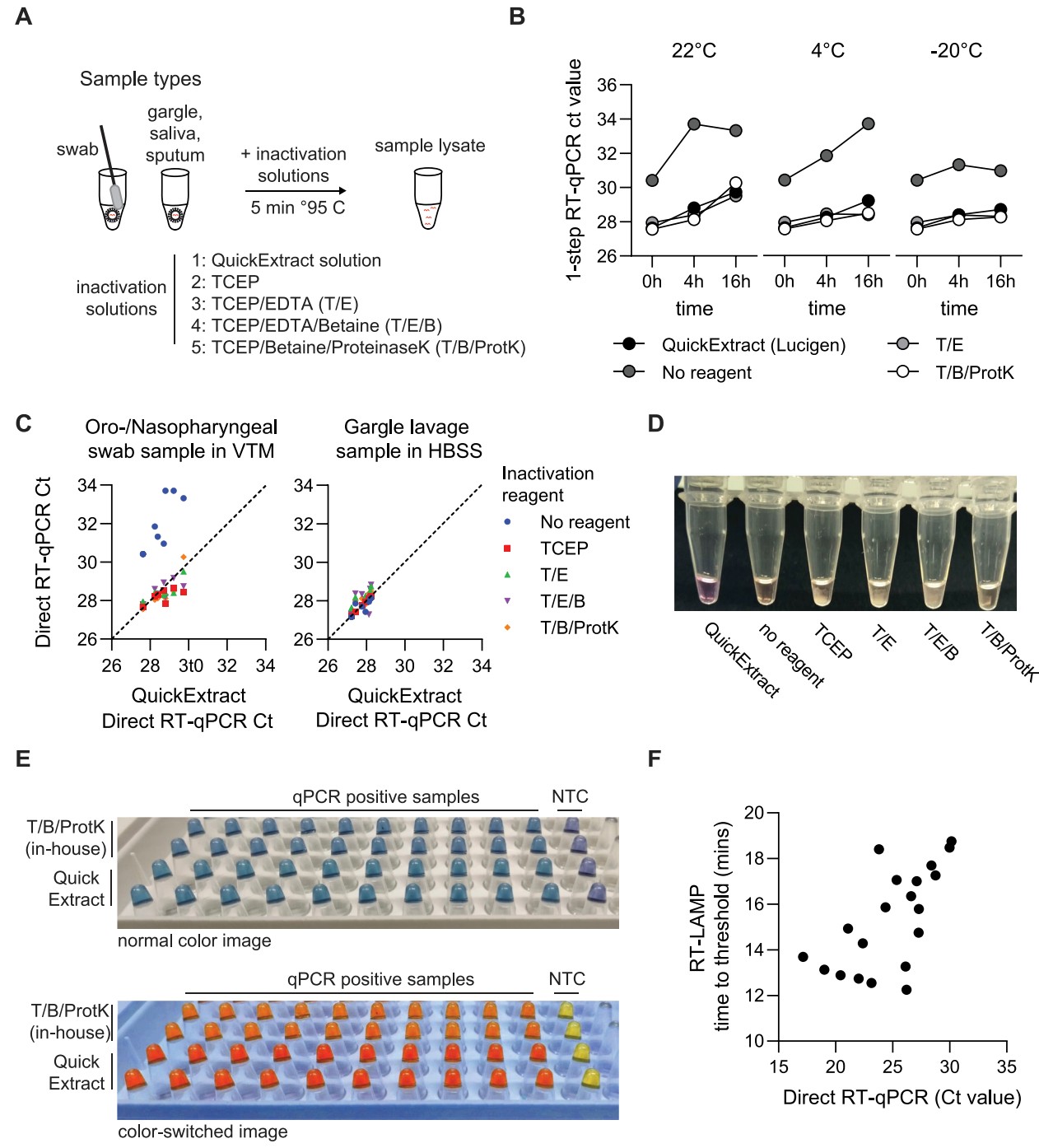

**Figure 2. Optimized, quick inactivation solution for direct-input RT-LAMP and RT–qPCR.**
**(A)** Scheme illustrating quick sample inactivation. Patient-derived (self- or professionally collected) respiratory sample is inactivated by the addition of an inactivation solution and heating for 5 min at 95°C. **(B)** Sample stability over time with different inactivation reagents. A mock sample in viral transfer medium was inactivated by adding inactivation reagents and heating for 5 min at 95°C. Samples were tested with one-step direct-input RT–qPCR at 0, 4, and 16 h of storage at RT (left), 4°C (middle), and −20°C (right). **(C)** Comparison of two commonly used respiratory tract samples. Sample stability after inactivation with different solutions was determined for independent swab samples in VTM or gargle lavage in HBSS. Ct values of open-source inactivation solutions were compared with QuickExtract solution. **(D)** Image demonstrating turbidity generated upon heat inactivation of VTM samples with different inactivation solutions. Smartphone image of mock nasopharyngeal sample in VTM taken after the addition of inactivation solutions and 5-min incubation at 95°C. **(E)** Colorimetry images of RT-LAMP reactions prepared after inactivation of samples with QuickExtract or in-house inactivation solution. Inactivated nasopharyngeal swab and gargle samples were tested by open-access RT-LAMP to assess the HNB colorimetric readout. **(F)** Comparison of time-to-threshold values for RT-LAMP and cycle-to-threshold values for RT–qPCR using direct-input samples. Time to threshold in minutes is reported for RT-LAMP, whereas cycle to threshold is reported for RT–qPCR. As evident from the poor correlation, quantitative measurement of target copy number is not possible via RT-LAMP.
Source data are available for this figure.

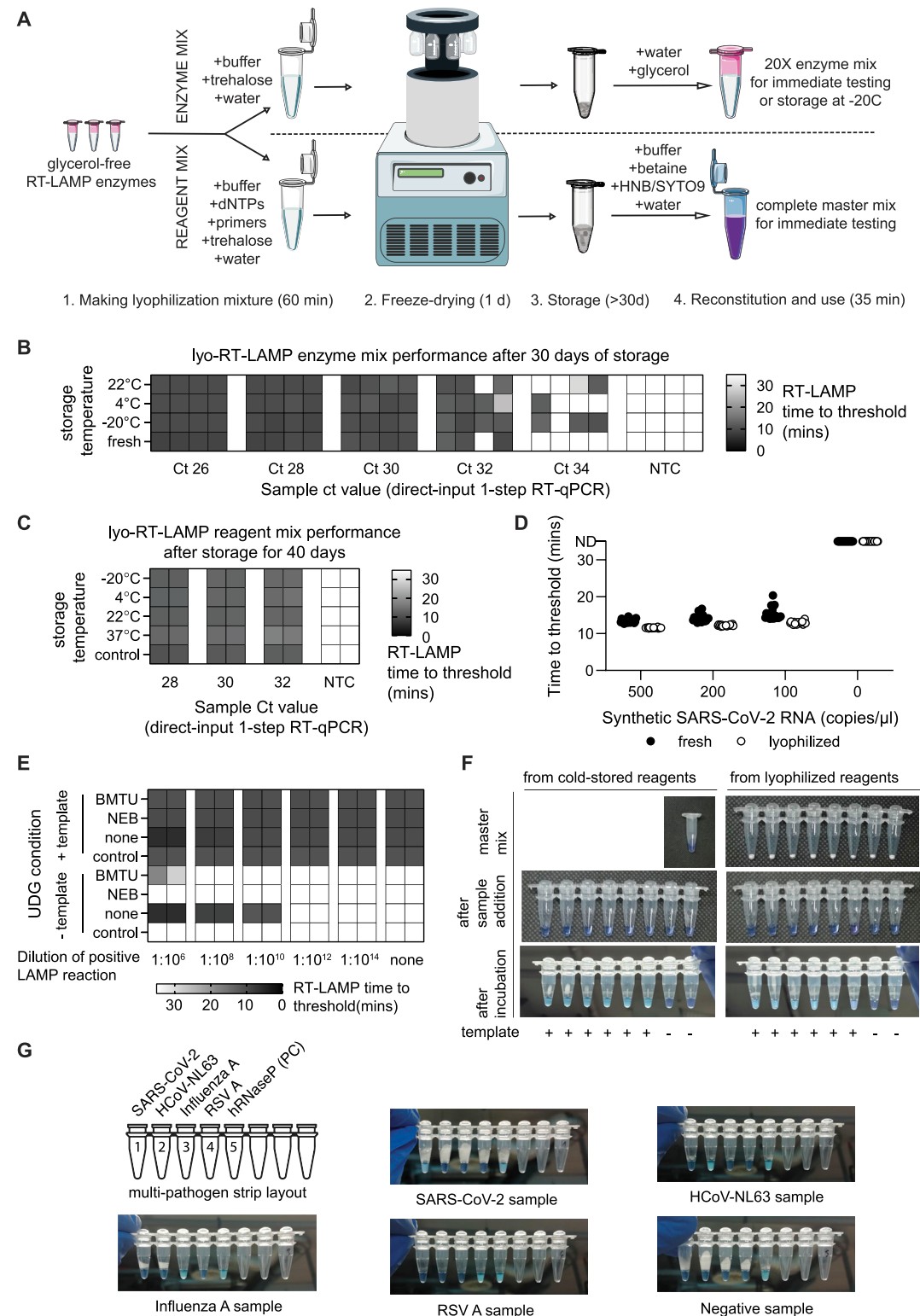

**Figure 3. Open-source lyophilized RT-LAMP mixture.**
**(A)** Schematic representation of lyophilization options for freeze-drying RT-LAMP enzymes and reagents. Glycerol-free enzymes can be lyophilized either as a concentrated enzyme mix or as a reagent mixture with dNTPs and primers. Trehalose is used in both options as a cryoprotectant. Mixtures are flash-frozen in liquid nitrogen, lyophilized using a freeze-dryer, and stored in dry conditions at various temperatures for transport. Before use, lyophilized mixtures are reconstituted.
**(B)** Performance of reactions assembled with reconstituted RT-LAMP enzyme mix. RT-LAMP enzyme mix was lyophilized and stored at 22°C, 4°C, and −20°C. After 30 d, lyophilized enzymes were reconstituted and used to assemble RT-LAMP reactions; control RT-LAMP reactions were prepared with nonlyophilized enzymes stored

strips were reconstituted by the addition of reconstitution buffer and tested on contrived samples containing individual virus RNA samples in a quick-inactivated negative control sample. The resulting visible color change revealed a clear and pathogen-specific nucleic acid amplification (Fig 3G), demonstrating the broad applicability of this simple solution for point-of-care testing. Overall, we have established two versatile lyophilization solutions for open-source RT-LAMP reagents that circumvent the current cold chain dependency of SARS-CoV-2 molecular test kit distribution and showed their potential to detect other viruses.

### Performance of direct-input open-source RT-LAMP on clinical samples

To determine the sensitivity and specificity of lyophilized open-source RT-LAMP reagents on clinical samples, two validation studies were conducted, one in Vienna, Austria, and another one in Accra, Ghana.

In Vienna, a total of 192 patient respiratory tract specimens (103 oro/nasopharyngeal swabs in VTM and 89 gargle samples in HBSS or saline solution [0.9% NaCl]) were heat-inactivated in homemade complete inactivation solution and tested with RT-LAMP using lyophilized open-access RT-LAMP reagent mix presented in this study. In parallel, samples were measured by direct-input RT–qPCR for reference and to obtain quantitative Ct values. 80 out of the 192 samples were identified as SARS-CoV-2–positive by direct-input RT–qPCR for the SARS-CoV-2 *N*-gene using the CDC N1 primer set. Direct-input lyophilized open-source RT-LAMP reactions showed a weak but detectable negative correlation of the concentration of viral RNA in patient samples with the time-to-threshold value for real-time fluorescence RT-LAMP reactions and resulted in robust HNB-based color changes for positive patient samples, indicating complete amplification after a run time of 35 min (Fig 4A and B). The lyo-RT-LAMP assay correctly detected 52 out of the 80 samples identified as positive by RT–qPCR (overall sensitivity of 65.0% [95% CI: 54.1–75.6%]), with a 50% limit of detection (LOD50) for a sample with a Ct value of 32.53 (95% C.I.: 31.34–33), corresponding to ~100 copies per reaction (Figs 4C and S4A). All 112 RT–qPCR–negative samples were

also identified as negative by lyo-RT-LAMP, giving a specificity of 100% (Figs 4D and S4B).

### Performance on clinical specimens at an independent laboratory in Ghana

To demonstrate the cold chain independence, robust deployability, and usability of our assay, we shipped lyophilized RT-LAMP reagent mixes to the West African Center for Cell Biology of Infectious Pathogens (WACCBIP) in Accra, Ghana (Fig 4E). A total of 200 respiratory tract (sputum) specimens with suspected SARS-CoV-2 infection were tested in this study. Samples were processed by a 5-min heat inactivation with homemade complete inactivation solution; in parallel, RNA extraction followed by RT–qPCR targeting the *N*-gene was performed as a reference. RT–qPCR identified 80 out of 200 samples as positive for SARS-CoV-2. Of these, 73 and 72 samples were detected as positive by real-time fluorescence and colorimetric lyophilized RT-LAMP, respectively (Fig 4F and G). The sensitivity of the assay on this sample set was 91.3% and 90.0% for direct-input samples, as measured by real-time fluorescence and colorimetry, respectively, with a $LOD_{50}$ for samples with a Ct value of 31.98 (95% C.I.: 30.65–33.57) (Fig 4H). The specificity for both sample input types and both readouts was 100%, as no false positives were detected in either fluorescence or colorimetry (Fig 4I). These data demonstrate that our established open-source lyophilized RT-LAMP reagent mix can be deployed at RT to remote locations with no apparent loss of reaction performance.

# Discussion

We developed a simple, cost-effective RT-LAMP system for nucleic acid detection using nonproprietary enzymes, tailored for point-of-care diagnostics in resource-limited settings. This work addresses two major barriers to equitable access in molecular testing: (1) the creation of an open-source, one-pot reagent mixture combining contamination-controlled single-step reverse transcription and isothermal amplification, and (2) the development of a lyophilized formulation that eliminates the need for cold chain transport and storage. Our colorimetric lyophilized RT-LAMP

---

at −20°C. Reactions were tested on a fourfold, five-step dilution series of a positive sample in four replicates by RT-LAMP, recording the time to threshold for each reaction. In addition, the sample dilutions were measured by direct-input RT–qPCR, and Ct values are displayed under the columns. **(C)** Time to threshold for reactions prepared using a reconstituted RT-LAMP reagent mix. RT-LAMP reagent mix was lyophilized and stored at −20°C, 4°C, 22°C, and 37°C. After 40 d, RT-LAMP reagents were reconstituted using the reconstitution buffer and reactions were tested on a fourfold, three-step dilution series of a positive sample in duplicates by RT-LAMP, recording the time to threshold for each reaction. In addition, the sample dilutions were measured by direct-input RT–qPCR, and Ct values are displayed under the columns. **(D)** Sensitivity comparison of RT-LAMP reactions assembled from lyophilized and nonlyophilized RT-LAMP reagents. Lyophilized and freshly prepared reactions were compared in 20 replicates using a dilution of SARS-CoV-2 synthetic RNA (Twist Biosciences) to assess the detection limit. Time-to-threshold values are shown. **(E)** Forced contamination experiment using lyophilized RT-LAMP reagents. In-house BMTU UDG enzyme, commercially available Antarctic Thermolabile UDG (New England Biolabs), and no UDG were included in three respective RT-LAMP reagent mixes for lyophilization. After freeze-drying, the reagent mixes were reconstituted into RT-LAMP master mixes and compared with freshly prepared reactions in a forced contamination experiment as described earlier (Fig 1C). **(F)** HNB colorimetric readout is compatible with lyophilized RT-LAMP reagent mixture. Reactions were prepared in parallel from cold-stored enzymes (left) and lyophilized reaction mix (right) to compare colorimetric readout. Smartphone images were taken after lyophilization/preparation of master mix (top row), after the addition of sample (middle row) and after a 35-min incubation at 63°C (bottom row) to showcase the color change. **(G)** Proof-of-concept multipathogen respiratory virus RT-LAMP test. An eight-well PCR strip was filled with singe-reaction aliquots of RT-LAMP reagent mix for different target RNAs and freeze-dried. A multipathogen test strip targeting SARS-CoV-2, human coronavirus NL63 (HCoV-NL63), influenza A, respiratory syncytial virus A (RSV A), and human RNase P (PC, positive control) was prepared from lyophilized reagents. Shown are the HNB RT-LAMP colorimetric results after reconstitution and addition of mock respiratory sample containing the respective pathogen RNA. A light blue color indicates a positive result.
Source data are available for this figure.

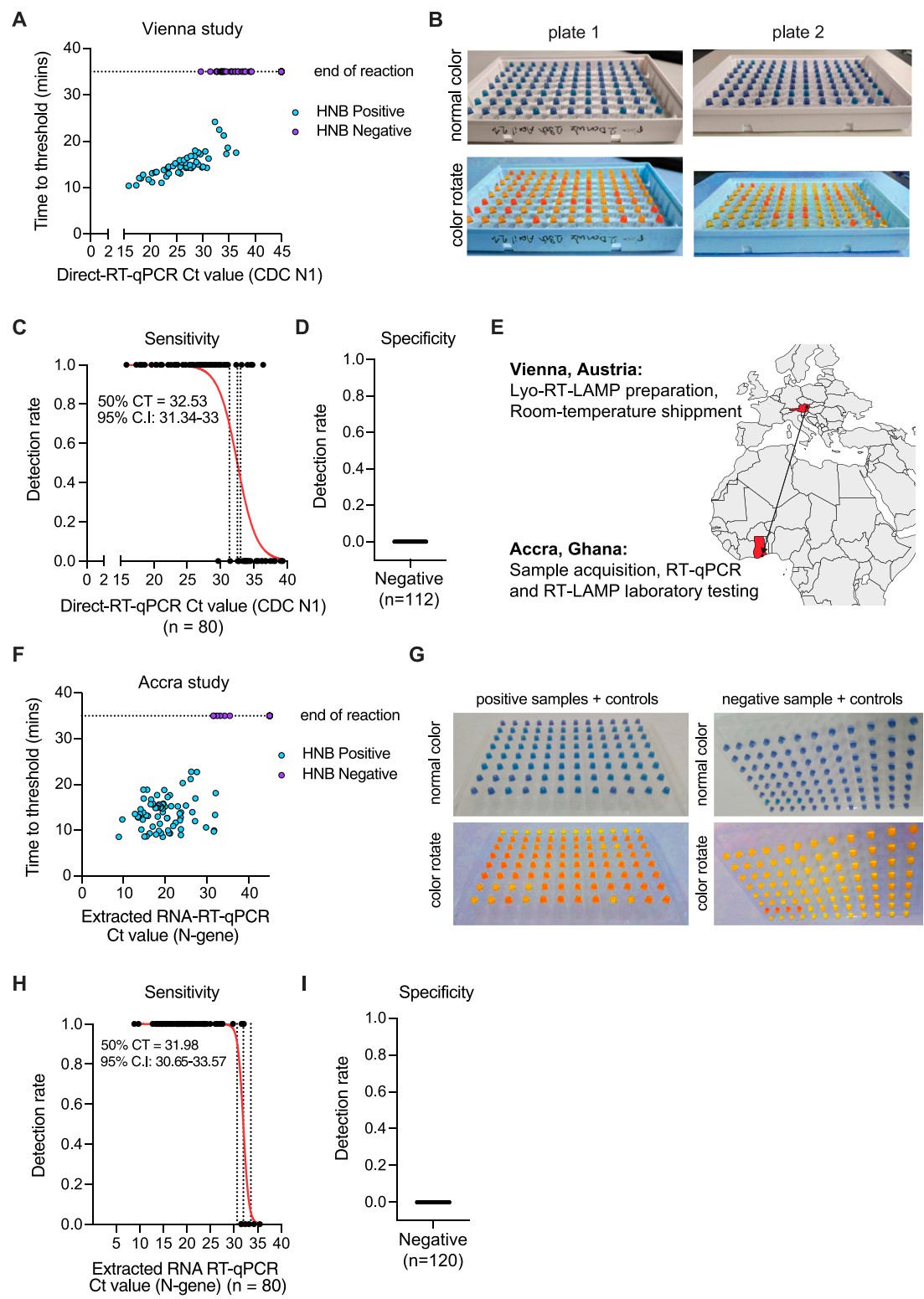

**Figure 4. Performance evaluation of lyophilized RT–LAMP reagent mix on clinical specimens in Austria and Ghana.**
**(A)** Performance of direct-input open-source RT–LAMP on clinical samples in Vienna. 192 samples with suspected COVID-19 infection were inactivated and tested in parallel with RT–qPCR targeting the *N*-gene, or HNB colorimetric RT–LAMP from lyophilized reagents. Shown are the time-to-threshold values obtained from real-time RT–LAMP reactions in comparison with RT–qPCR Ct values. Each dot represents an individual sample. The color of each dot indicates the colorimetric results from HNB RT–LAMP reactions. **(B)** Colorimetric HNB RT–LAMP results from samples shown in (A). Raw (top) or color-converted (bottom) images are shown. **(C)** Sensitivity of lyophilized RT–LAMP reactions. Each dot represents a single sample. A simple logistic regression analysis was performed for samples containing different amounts of viral RNA (measured via RT–qPCR) and plotted as a red line. 95% confidence intervals and 50% detection limits are indicated as dashed lines and added as a textbox.

assay performed comparably to commercial solutions, enabling sensitive detection across a range of environments and targets.

Central to our approach was the identification of nonproprietary enzyme components that provide robust, sensitive, and specific RT-LAMP–based detection of respiratory viruses, including but not limited to SARS-CoV-2. We found that *Bst* LF DNA polymerase displayed strong performance across various amplicons and sample matrices, including both extracted RNA and heat-inactivated crude samples. Although WT *Bst* LF is reported to be sensitive to high salt and dUTP concentrations, our assays maintained high performance even with saline swab specimens and at sufficient dUTP concentrations enabling contamination-controlled single-step reverse transcription. Nevertheless, alternatives such as the more thermophilic and urea-tolerant *Bst* LF variant from the Ellington laboratory (Paik et al, 2023) may offer improved robustness for challenging sample types.

Our one-pot system combines *Bst* LF with the *BH10* strain of HIV-1 reverse transcriptase and the thermolabile BMTU uracil-DNA glycosylase (Jaeger et al, 2000; Notomi et al, 2000; Álvarez et al, 2009). This combination enables both carry-over contamination prevention during reaction setup and efficient amplification at elevated assay temperatures. We also optimized a simple inactivation buffer for crude sample processing, improving on earlier protocols by incorporating betaine and proteinase K (Myhrvold et al, 2018; Rabe & Cepko, 2020). This formulation supported stable performance even after extended storage of inactivated samples, a key requirement for decentralized testing workflows.

Although our protocols provide a path to affordable and open-access molecular diagnostics, we acknowledge that reaction performance remains sensitive to reagent quality and requires optimization. Homemade RT-LAMP systems can be inconsistent when enzyme production or purification is suboptimal. This is a notable disadvantage compared with commercial kits, particularly for laboratories without access to protein purification tools such as FPLC systems used in our study. In such cases, partnerships with equipped facilities or adoption of simplified purification methods may offer viable alternatives (de Souza et al, 2023). However, inconsistent enzyme quality and low purity can reduce diagnostic reliability. When optimizing homemade RT-LAMP assays, establishing standardized protocols and control processes is equally critical. Long-term solutions should therefore also prioritize building local manufacturing infrastructure for enzyme production and quality assurance, to support sustainable and equitable testing capacity (Banda et al, 2021).

Lyophilization of reagents addresses the logistical hurdle of cold chain dependency in molecular diagnostics (Talbot et al, 2025). Although others have reported lyophilized RT-LAMP systems (Carter et al, 2017; Howson et al, 2017), we developed and validated two complementary protocols: one for enzyme-only mixes (*Bst* LF, HIV-RT, BMTU UDG) and one for complete reactions that also included dNTPs and primers. Both retained the visual colorimetric readout of hydroxynaphthol blue (HNB), which is essential for instrument-free diagnostics. Further studies are required to assess long-term stability (e.g., 2 yr), as has been achieved by leading commercial products (e.g., *LyoPrime Warm-Start RT-LAMP Mix* by New England Biolabs or TB LAMP by Eiken Chemical).

To support real-world adoption, we envision several implementation models, ranging from centralized production and distribution of lyophilized reagents to decentralized in-lab preparation by regional facilities. Broader factors such as infrastructure, regulatory frameworks, and supply chain logistics will significantly influence deployment and long-term sustainability. Although a detailed exploration of these factors is beyond the scope of this study, our work addresses key technical challenges that currently limit the scalability of decentralized diagnostics. By validating the system's performance across diverse sample types and international settings, we demonstrate that the platform is both robust and suitable for field implementation.

In conclusion, we present an open-source lyophilized RT-LAMP solution suitable for decentralized colorimetric nucleic acid testing in resource-limited settings. Our protocols empower laboratories and public health institutions to produce and use diagnostic reagents largely independent of commercial supply chains. This is especially needed for outbreak scenarios that require rapid responses and implementation of test–trace–isolate containment strategies (Larremore et al, 2021). Our insights gained during the COVID-19 pandemic are shared via rtlamp.org and include practical guidance on contamination control, assay readout, and implementation of open-source colorimetric RT-LAMP beyond SARS-CoV-2. This work contributes to building global diagnostic equity by enabling accessible, reliable pathogen surveillance and outbreak response tools.

# Materials and Methods

### Expression and purification of open-access RT-LAMP enzymes

Expression plasmids for *Bst* LF, HIV-1 RT, and BMTU UDG were prepared by Gibson cloning of synthetic DNA containing the coding sequences of the respective enzymes (gBlocks, IDT) into pET-based

---

**(D)** Specificity of lyophilized RT-LAMP reactions. Each dot represents a single sample. RT-LAMP detection rates for true-negative samples determined by RT–qPCR are shown. **(E)** Clinical sample validation performed in Vienna, Austria, and Accra, Ghana. Lyophilized reagents were prepared in Vienna and shipped to Accra. Each site performed an independent validation study. **(F)** Performance of direct-input open-source RT-LAMP on clinical samples in Accra, Ghana. RNA from 192 samples with suspected COVID-19 infection was extracted and tested in parallel with RT–qPCR targeting the *N*-gene, or HNB colorimetric RT-LAMP from lyophilized reagents. Shown are the time-to-threshold values obtained from real-time RT-LAMP reactions in comparison with RT–qPCR Ct values. Each dot represents an individual sample. The color of each dot indicates the colorimetric results from HNB RT-LAMP reactions. **(G)** Colorimetric HNB RT-LAMP results from samples shown in (F). Raw (top) or color-converted (bottom) images are shown. **(H)** Sensitivity of lyophilized RT-LAMP reactions. Each dot represents a single sample. A simple logistic regression analysis was performed for samples containing different amounts of viral RNA (measured via RT–qPCR) and plotted as a red line. 95% confidence intervals and 50% detection limits are indicated as dashed lines and added as a textbox. **(I)** Specificity of lyophilized RT-LAMP reactions. Each dot represents a single sample. RT-LAMP detection rates for true-negative samples determined by RT–qPCR are shown.
Source data are available for this figure.

expression vectors. Expression plasmids are available through Addgene: plasmid IDs 159148 (*Bst* LF), 159149 (HIV-1 RT), and 172197 (BMTU UDG). *E. coli* BL21 (DE3) bacterial cells were transformed with the respective expression plasmids, and overnight cultures were grown at 37°C in LB medium at 180 rpm in the presence of ampicillin. Large-scale expression was performed in the ZYP-5052 autoinduction medium (Studier, 2005). The cells were grown for 5 h 30 min at 37°C, followed by overnight expression at 18°C. The cells were then harvested by centrifugation (4,000*g*, 15 min) and stored at –80°C.

*Bst* LF purification was carried out as follows: the cells were resuspended in the lysis buffer (20 mM Tris–HCl, pH 7.5, 500 mM NaCl, 0.5 mM TCEP, and Benzonase). Cell lysis was accomplished using a single cycle in a cell disruptor (Constant Systems Ltd.), with the pressure set to 1.4 kBar. Lysates were clarified by centrifugation (42,000*g*, 45 min, 4°C), and a two-step purification protocol using an ÄKTA Protein Purification System (GE Healthcare Life Sciences) at 8°C was then followed to obtain the recombinant protein. First, the supernatant was applied to the His-Trap Crude 5-ml column (GE Healthcare Life Sciences) previously equilibrated with the lysis buffer. The column was washed with 20 column volumes of the lysis buffer, containing 20 mM imidazole. Then, the protein was eluted using a step gradient of imidazole (equilibration buffer with 250 mM imidazole and then with 500 mM imidazole). Fractions were examined by SDS–PAGE for protein content and purity and pooled according to the presence of *Bst* LF and diluted with the RQA buffer containing 20 mM Tris–HCl, pH 7.5, 0.5 mM TCEP to a final concentration of 50 mM NaCl. As a second step, the diluted fractions were applied to a Resource Q 6-ml ion exchange column (GE Healthcare Life Sciences) pre-equilibrated with the RQA buffer. The protein was then eluted using a linear gradient of 0.05–0.74 M NaCl in the RQA buffer. The peak fractions were combined and dialyzed overnight against storage buffer consisting of 40 mM Tris–HCl, pH 7.5, 50 mM NaCl, 1 mM DTT, and 10% glycerol, flash-frozen in liquid nitrogen, and stored at –80°C until use.

HIV-1 RT purification was carried out as follows: the cells were resuspended in the lysis buffer (20 mM Tris–HCl, pH 7.5, 500 mM NaCl, 0.5 mM TCEP, and Benzonase). Cell lysis was accomplished using a single cycle in a cell disruptor (Constant Systems Ltd.), with the pressure set to 1.4 kBar. Lysates were clarified by centrifugation (42,000*g*, 45 min, 4°C), and a three-step purification protocol using an ÄKTA Protein Purification System (GE Healthcare Life Sciences) at 8°C was then followed to obtain the recombinant protein. First, the supernatant was applied to the His-Trap Crude 5-ml column (GE Healthcare Life Sciences) previously equilibrated with the lysis buffer. The column was washed with 20 column volumes of the lysis buffer, containing 20 mM imidazole. Then, the protein was eluted using a step gradient of imidazole (equilibration buffer with 250 mM imidazole and then with 500 mM imidazole). Fractions were examined by SDS–PAGE for protein content and purity and pooled according to the presence of HIV-1 RT and diluted with the RSA buffer containing 20 mM Tris–HCl, pH 7.5, 0.5 mM TCEP to a final concentration of 50 mM NaCl. As the second step, diluted fractions were applied to the Resource S 6-ml ion exchange column (GE Healthcare Life Sciences) pre-equilibrated with the RSA buffer. The protein was then eluted using a linear gradient of 0.05–0.74 M NaCl in the RQA buffer. For the third step of the purification protocol,

peak fractions were combined and applied to a HiLoad 16/600 Superdex 200 pg column (GE Healthcare Life Sciences) equilibrated with the storage buffer consisting of 50 mM Tris–HCl, pH 7.5, 50 mM NaCl, 0.5 mM TCEP, and 10% glycerol. Eluted fractions were analyzed by SDS–PAGE, pooled according to the protein purity, concentrated, flash-frozen in liquid nitrogen, and stored at –80°C until use.

BMTU 3346 UDG purification was carried out as follows: the cells were resuspended in the lysis buffer (20 mM Tris–HCl, pH 8.0, 500 mM NaCl, 0.5 mM TCEP, and Benzonase). Cell lysis was accomplished using a single cycle in a cell disruptor (Constant Systems Ltd.), with the pressure set to 1.4 kBar. Lysates were clarified by centrifugation (42,000*g*, 45 min, 4°C), and a two-step purification protocol using an ÄKTA Protein Purification System (GE Healthcare Life Sciences) at 8°C was then followed to obtain the recombinant protein. The supernatant was applied to the His-Trap Crude 5-ml column (GE Healthcare Life Sciences) previously equilibrated with the lysis buffer. The column was washed with 20 column volumes of the lysis buffer, containing 20 mM imidazole. Then, the protein was eluted using a step gradient of imidazole (equilibration buffer with 250 mM imidazole and then with 500 mM imidazole). Fractions were examined by SDS–PAGE for protein content and purity and pooled according to the presence of BMTU UDG and diluted with the RQA buffer containing 20 mM Tris–HCl, pH 8.0, 0.5 mM TCEP to a final concentration of 50 mM NaCl. Diluted fractions were applied to the Resource Q 6-ml ion exchange column (GE Healthcare Life Sciences) pre-equilibrated with the RQA buffer. The protein was then eluted using a linear gradient of 0.05–0.74 M NaCl in the RQA buffer. The peak fractions were combined and dialyzed overnight against storage buffer consisting of 20 mM Tris–HCl, pH 8.0, 50 mM NaCl, 0.5 mM TCEP, flash-frozen in liquid nitrogen, and stored at –80°C until use.

### Assembly of RT-LAMP reactions to identify open-source enzymes

Reactions comparing different RT-LAMP enzymes were carried out in 1X Isothermal Amplification Buffer (20 mM Tris–HCl, pH 8.8, 10 mM $(NH_4)_2SO_4$, 50 mM KCl, 2 mM $MgSO_4$, 0.1% Tween-20), 1.4 mM dNTP mix (25 mM stock, Larova), 0.7 mM dUTP (100 mM stock, Larova), 6 mM $MgSO_4$ (8 mM final including magnesium sulfate from buffer, NEB), 0.4 M betaine (Sigma-Aldrich), 2 $\mu$M SYTO 9 fluorescent dye (100 $\mu$M stock in DMSO; Thermo Fisher Scientific), 120 $\mu$M hydroxynaphthol blue trisodium salt (from 3 mM stock in nuclease-free water, Hach), oligonucleotide primers from the HMS-1 primer set (Rabe & Cepko, 2020) at final concentrations of 0.2 $\mu$M for F3/B3, 0.4 $\mu$M for LB/LF, and 1.6 $\mu$M FIP/BIP (Sigma-Aldrich), and nuclease-free water to a total volume of 8 $\mu$l, reserving 2 $\mu$l for sample volume and volume for enzymes (see below).

To test different DNA polymerases, WarmStart RTx reverse transcriptase (NEB) was added at 0.3 U/$\mu$l and Antarctic Thermolabile UDG (NEB) was added at 0.02 U/$\mu$l to all reactions. DNA polymerases were added to individual reactions at final concentrations according to the manufacturer's instructions, namely, 0.32 U/$\mu$l *Bst* DNA polymerase large fragment (NEB), 0.32 U/$\mu$l *Bst* 2.0 DNA polymerase (NEB), 0.32 U/$\mu$l *Bst* 2.0 WarmStart DNA polymerase (NEB), 0.32 U/$\mu$l *Bst* 3.0 DNA polymerase (NEB), whereas

*Bst* LF DNA polymerase (in-house–produced) was used at the empirically determined optimal concentration of 20 ng/μl.

To test different reverse transcriptases, in-house *Bst* LF DNA polymerase was added at a final concentration of 20 ng/μl and Antarctic Thermolabile UDG (NEB) was added at 0.02 U/μl to all reactions. Reverse transcriptases were added according to the manufacturer's instructions at the final concentrations of 0.2 U/μl AMV reverse transcriptase (NEB), 4 U/μl M-MuLV reverse transcriptase (NEB), 0.3 U/μl WarmStart RTx reverse transcriptase (NEB), whereas HIV-1 reverse transcriptase (in-house–produced) was used at the empirically determined concentration of 7.5 ng/μl.

Reactions with different DNA polymerases and reverse transcriptases were tested by the addition of synthetic SARS-CoV-2 RNA (Twist Biosciences) diluted in nuclease-free water to a concentration of 1,000, 500, 250, 50, 25, and 0 copies/μl in the sample and incubation at 63°C for 60 min, with real-time fluorescence reading in SYBR/FAM wavelength every minute in a Bio-Rad CFX96 instrument. Reactions were performed in four replicates.

To test different uracil-DNA glycosylase enzymes, reactions were prepared as stated above, differing in the enzymes and primers used. LAMP ORF3a-A primer set from Schermer et al (2020) was used exclusively in this experiment to avoid contamination for the primer pair used in the rest of the study. In-house *Bst* LF DNA polymerase and in-house HIV-1 reverse transcriptase were added at a final concentration of 20 and 7.5 ng/μl, respectively, to all reactions. A precontamination reaction was set up with the LAMP ORF3a-A primer set, and a SARS-CoV-2–positive sample was amplified by incubation at 63°C for 30 min. The resulting positive reaction was serially diluted in a 1:100 manner, diluting the positive reaction to 1:10$^2$, 1:10$^4$, 1:10$^6$, 1:10$^8$, 1:10$^{10}$, respectively. RT-LAMP master mixes were prepared with LAMP ORF3a-A primer set and HIV-1 RT and *Bst* LF DNA polymerase as described above. For the commercial UDG condition, Antarctic Thermolabile UDG (NEB) was added to the reactions in the final concentration of 0.02 U/μl; for the BMTU UDG condition, in-house–purified BMTU UDG was added to the reactions in the final concentration of 0.025 ng/μl; and for the no UDG condition, UDG storage buffer was added to the reactions to match the volume of added UDG in the previous conditions. Positive inactivated SARS-CoV-2 sample was added to half of the reactions (+ template), and negative sample was added to the second half (– template), to distinguish between true positive and false positive in the presence of amplicons. Reactions were spiked with the contaminant dilutions and incubated at 63°C for 35 min. The reactions were performed in duplicates.

To test the thermostability of UDG enzymes using a qPCR-based assay, two dsDNA template amplicons were prepared: (i) a 0.4-kb dsDNA amplicon generated via PCR with equimolar ratios of dTTP and dUTP in the reaction and (ii) 1-kb dsDNA amplicon generated via PCR with no dUTP present in the reaction. These are the dUTP and dTTP target, respectively. Two separate PCR master mixes were prepped to amplify the two respective targets, the reaction consisting of 1X Standard Taq Reaction Buffer (NEB), 250 μM dNTPs each (including dUTP), 0.1 μM forward primer, 0.1 μM reverse primer, 1 μM SYTO 9 intercalating fluorescent dye, 0.25 U *Taq* DNA polymerase, and template dsDNA (0.01 ng/μl) to a final reaction volume of 20 μl, reserving 0.4 μl for the addition of UDG enzyme. In-

house–expressed/purified BMTU UDG enzyme (2.5 ng/μl) and Antarctic Thermolabile UDG (NEB) were incubated in a thermocycler on a gradient setting (30°C–65°C) for 5 min. After incubation, UDG enzymes were immediately put on ice and 0.4 μl of enzyme was added to each PCR on ice. PCR was performed in a Bio-Rad CFX qPCR cycler with the following thermal cycling settings: denaturation at 96° for 30 s, annealing at 60°C for 30 s, elongation at 72°C for 60 s, repeated for a total of 45 cycles. Fluorescence readings in the FAM/SYBR range were taken after every elongation cycle.

### Open-source RT-LAMP and bead-LAMP assays

RT-LAMP master mix consists of 1X Isothermal Amplification Buffer (20 mM Tris–HCl, pH 8.8, 10 mM (NH$_4$)$_2$SO$_4$, 50 mM KCl, 2 mM MgSO$_4$, 0.1% Tween-20, entire buffer prepared from stock solutions as a 10X concentrate or purchased from NEB, stored at –20°C), 1.4 mM dNTPs each (25 mM each, stock solution in ultrapure water, Larova), 0.7 mM dUTP (100 mM stock in ultrapure water; Larova), 6 mM MgSO$_4$ (8 mM final including magnesium sulfate from Isothermal Amplification Buffer, from 1 M stock solution or 100 mM stock solution NEB), 0.4 M betaine (from 5 M stock solution prepared from powder, solution stored at –20°C; Sigma-Aldrich), 2 μM SYTO 9 fluorescent dye (100 μM stock solution in DMSO, for performing fluorescence readout; Thermo Fisher Scientific), 120 μM hydroxynaphthol blue (HNB) trisodium salt (from 3 mM stock in nuclease-free water, stored at 4°C or –20°C; Hach), oligonucleotide primers from the HMS-1 primer set for SARS-CoV-2 detection (Rabe & Cepko, 2020) at final concentrations of 0.2 μM for F3/B3, 0.4 μM for LB/LF, and 1.6 μM FIP/BIP (resuspended in nuclease-free water; Sigma-Aldrich), 20 ng/μl *Bst* LF DNA polymerase, 7.5 ng/μl HIV-1 reverse transcriptase, 0.025 ng/μl BMTU UDG, and nuclease-free water to a volume of 16 μl, reserving 4 μl for sample input volume to a total of 20 μl final reaction volume. Inactivated crude samples or extracted RNA is added, and the reaction is mixed by pipetting up and down several times. After that, reaction vessels were capped or sealed and incubated at 63°C for 35 min. For real-time fluorescence detection, reactions were prepared in optically clear PCR plates and optically clear plate seals were used. Reactions were incubated in a real-time thermocycler at a block temperature of 63°C, with lid temperature exceeding that of the block by at least 5°C and with real-time fluorescence readings taken every 60 s in the SYBR/FAM filter to capture SYTO 9 fluorescence. Colorimetric results of the bead-LAMP assays were evaluated immediately after incubation, and finished reactions were discarded to avoid cross-contamination risk.

Bead-LAMP reaction mix was prepared as above with double the amount of *Bst* LF DNA polymerase (40 ng/μl final concentration) omitting volume left for sample and replacing it with nuclease-free water to a total volume of 20 μl reaction mix per reaction. Bead extraction was performed in PCR strips or 96-well PCR plates in the following way. In-house–prepared silica-coated magnetic beads for nucleic acid extraction (Oberacker et al, 2019) were diluted 1:5 in a bead dilution buffer (2.5 M NaCl, 10 mM Tris–HCl, pH 8.0, 20% [wt/vol] PEG 8000, 0.05% Tween-20). 100 μl of inactivated sample or extracted RNA was mixed with 60 μl of the diluted beads and pipetted up and down several times to mix completely. Bead–sample mixture was incubated at RT for 5 min to allow for

binding of nucleic acids to the magnetic beads. Beads were then separated from the solution by placing them on a magnet for 5 min at RT. After that, the solution was removed and beads were washed twice by adding 150 $\mu$l of 85% ethanol over the beads and removing it after no longer than 30 s, keeping the samples on the magnet the whole time. Remaining ethanol was removed, and beads were left to air-dry for a maximum of 3 min to avoid overdrying. 20 $\mu$l of bead-LAMP reaction mix was directly added to the beads, and beads were resuspended in the mix by pipetting up and down or sealing the plate and gently vortexing. Sealed or capped reaction vessels were incubated at 63°C for 35 min. For real-time fluorescence detection, reactions were prepared in optically clear PCR plates and optically clear plate seals were used. Reactions were incubated in a real-time thermocycler at a block temperature of 63°C, with lid temperature exceeding that of the block by at least 5°C and with real-time fluorescence readings being taken every minute in the SYBR/FAM filter to capture SYTO 9 fluorescence. Colorimetric results of the bead-LAMP assays were evaluated immediately after incubation, and reactions were discarded.

## Quick inactivation of sample material to bypass RNA extraction

Naso/oropharyngeal swabs collected in viral transport medium (VTM) and gargle samples collected in 5 ml 0.9% NaCl solution (saline) with roughly the same viral load were selected for RNA stability over time experiment. The samples were heat-inactivated by incubation at 95°C for 5 min after the addition of an inactivation reagent. The tested inactivation reagents were as follows: QuickExtract DNA Extraction Solution (used as 2X solution); no reagent; 25 mM Tris (2-carboxyethyl)phosphine hydrochloride solution, pH 7.0 (TCEP, used as 10X solution); 25 mM TCEP, 10 mM EDTA (TCEP/EDTA, used as 10X solution); 25 mM TCEP, 10 mM EDTA, 4 M betaine (T/E/B, used as 10X solution); complete inactivation solution: 25 mM TCEP, 10 mM EDTA, 4 M betaine, 2 mg/ml proteinase K (used as 10X solution). Samples were put on ice and immediately measured with direct-input RT-qPCR in two technical duplicates as per Kellner et al (2021) Preprint using the CDC N1 assay. Samples were split into three groups and stored at RT, 4°C, and −20°C. Samples were tested after four and 16 h post-inactivation with direct-input RT–qPCR as described above. Delta Ct values were calculated from the geometric mean of the technical duplicates and plotted for each inactivation reagent into separate plots consisting of temperature and biological sample condition.

To demonstrate wide sample compatibility with the complete homemade and QuickExtract inactivation reagents, 10 SARS-CoV-2–positive nasopharyngeal swabs collected in VTM and 10 SARS-CoV-2–positive gargle samples collected in saline were inactivated with QuickExtract (used as 2X solution) and complete homemade (used as 10X solution) by incubation at 95°C for 5 min. The samples were placed on ice and immediately tested via direct-input RT–qPCR (as described in Kellner et al [2022]) to quantitatively assess the viral load, and the Ct values were compared with those obtained from these same samples with the gold-standard diagnostic RNA-extracted RT–qPCR method performed at the Austrian Agency for Health and Food Safety (AGES).

## Lyophilization of open-source RT-LAMP enzymes

Open-source RT-LAMP enzymes (*Bst* LF, HIV-1 RT, and BMTU UDG) were expressed and purified as stated in this study, with a final storage buffer excluding glycerol. Protein concentration was determined using DeNovix DS-11 FX + Spectrophotometer/Fluorometer using calculated extinction coefficients and molecular weight values and confirmed via SDS–PAGE gel electrophoresis.

To prepare freeze-dried RT-LAMP enzyme mix, an enzyme lyophilization mixture consisting of 1x Isothermal Amplification Buffer (20 mM Tris–HCl, 10 mM $(NH_4)_2SO_4$, 50 mM KCl, 2 mM $MgSO_4$, 0.1% Tween-20), 10% trehalose (wt/vol), 0.8 $\mu$g/$\mu$l *Bst* LF, and 0.3 $\mu$g/$\mu$l HIV-1 RT was prepared on ice, aliquoted into 1.5-ml Eppendorf tubes with pierced caps, and flash-frozen in liquid nitrogen for 5 min. Tubes were immediately placed in a Christ Alpha 2-4 LDplus freeze-dryer set to the main drying program with chamber temperature at −78°C and pressure at 0.091 mbar. Tubes with lyophilized enzyme mixtures were collected after 16 h of freeze-drying, capped with unpierced, normal lids, and stored in sealable plastic bags with silica gel desiccant packets to protect the contents of the tubes from atmospheric moisture during storage.

To prepare freeze-dried LAMP reagents, a lyophilization mix was prepared consisting of 0.8X Isothermal Amplification Buffer (prepared as above, excluding $MgSO_4$), 7 mM dATP, dUTP, dTTP, and dCTP each, 3.5 mM dUTP, 1 $\mu$M F3/B3 primers, 2 $\mu$M LF/LB primers, and 8 $\mu$M FIP/BIP primers, 100 ng/$\mu$l *Bst* LF DNA polymerase, 37.5 ng/$\mu$l HIV-1 RT, 0.125 ng/$\mu$l BMTU 3346 UDG, and 10% trehalose, adding nuclease-free water to a final volume of 4 $\mu$l lyophilization mix per 20 $\mu$l reaction. This mixture was lyophilized and stored in the same way as detailed above for RT-LAMP enzymes.

Freeze-dried enzyme mixes were reconstituted by adding an amount of nuclease-free water to reach the volume of the lyophilized mix before freeze-drying. The white pellet dissolved immediately forming a ready-to-use 20X RT-LAMP enzyme mix. If storage of reconstituted enzyme mixes at −20°C is desired, reconstitution is performed with 10% glycerol in nuclease-free water instead. Reactions with reconstituted enzymes were assembled as normal RT-LAMP reactions with reconstituted enzymes being added at 1X final concentration instead of normal enzyme addition. Stability of freeze-dried LAMP enzyme mixes was tested by lyophilizing several aliquots of enzyme mix and storing them at −20°C, 4°C, and RT to assess temperature-dependent stability. RT-LAMP reactions were assembled as described above after 0, 10, and 30 d after lyophilizing of enzyme mixes. Reactions were tested for sensitivity on a dilution series of SARS-CoV-2–positive samples with reactions assembled with nonlyophilized enzymes as a control group. Reactions were performed in four replicates.

Freeze-dried RT-LAMP reagent mixes were reconstituted with the addition of a reconstitution buffer that is composed of nuclease-free water, 1X Isothermal Amplification Buffer, 8 mM MgSO4, 0.4 M betaine, 120 $\mu$M HNB dye, and 2 $\mu$M SYTO 9 fluorescent dye. The white pellet was readily dissolved, and the resultant mixture formed a complete RT-LAMP reaction mix that was distributed among PCR strips or PCR plates, dispensing 16 $\mu$l per well. After this, 4 $\mu$l inactivated sample was added and mixed by

pipetting up and down and reactions were incubated at 63°C for 35 min. Stability of freeze-dried RT-LAMP reagent mixes was tested by lyophilizing several aliquots of RT-LAMP reaction mix and storing them at −20°C, 4°C, RT, and 37°C to assess temperature-dependent stability. RT-LAMP reactions were assembled as described above after 0, 10, and 30 d after lyophilizing of enzyme mixes. Reactions were tested for sensitivity on a dilution series of SARS-CoV-2–positive samples with reactions assembled with nonlyophilized enzymes as a control group. Reactions were performed in duplicates.

### Clinical sample collection and ethics

Patient samples in Vienna (oro/nasopharyngeal swabs and gargle) were obtained as part of a clinical performance study approved by the local Ethics Committee of the City of Vienna (#EK 20-292-1120). The patients/participants provided their written informed consent to participate in this study. Oro/nasopharyngeal swabs were collected in 3 ml viral transport medium (VTM) or 0.9% NaCl solution (saline). Gargle samples were collected from patients by letting individuals gargle for 1 min with 5 ml of HBSS or 0.9% saline solution. Informed consent was obtained from all patients.

For sample collection in Ghana, an ethical approval was obtained from the Ghana Health Service Ethics Review Committee (GHS-ERC 011/03/20). Sputum samples were obtained as part of a larger COVID-19 surveillance study from individuals who tested for COVID-19 at the West African Centre for Cell Biology of Infectious Pathogens (WACCBIP), University of Ghana, during the pandemic (2021). A written informed consent was obtained from all participants before enrolment. The sputum samples were obtained in 50 ml Falcon tubes and stored at −80°C until ready for use.

### Lyophilized open-source RT-LAMP validation in Vienna, Austria

A total of 192 de-identified respiratory tract specimens were obtained from the Austrian Agency for Health and Food Safety and heat-inactivated using the homemade complete inactivation solution. Samples were stored at −80°C before use.

Direct RT–qPCR was performed using the Luna Universal One-Step RT–qPCR kit (NEB), 1.5 $\mu$l of reference primer/probe set CDC N1 (IDT), 0.4 $\mu$l of Antarctic Thermolabile UDG (NEB), and 2 $\mu$l of inactivated sample per 20 $\mu$l reaction. PCR program included reverse transcription at 55°C for 10 min, initial denaturation at 95°C for 1 min followed by 45 cycles of 95°C for 10 s and 55°C for 30 s in a Bio-Rad CFX qPCR cycler. For inferring viral load from direct RT–qPCR Ct values, SARS-CoV-2 RNA Synthetic Control 2 (1,000,000 copies/$\mu$l, Twist Bioscience) was diluted in heat-inactivated 0.9% NaCl. 10,000, 1,000, and 100 copies/$\mu$l dilutions were performed and tested by the protocol above in duplicates. The mean Ct values were linearized and used to construct an equation via linear regression (Fig S4A), which was used to relate Ct values into copies/$\mu$l. Lyophilized open-source RT-LAMP mix was prepared as stated above and reconstituted for use in the experiment immediately after lyophilization. Real-time fluorescence detection, as well as colorimetric detection after 35-min incubation, was performed, and the colorimetric results were scored

by the researcher immediately after incubation using the colorimetry.net web app.

### Lyophilized open-source RT-LAMP validation in Accra, Ghana

A total of 200 cryopreserved sputum samples (80 SARS-CoV-2 positives and 120 SARS-CoV-2 negatives) were selected from a previously analyzed sample pool. RNA was purified from 180 $\mu$l of sample using QIAamp Viral RNA Mini Kit (QIAGEN) following the instructions from the manufacturer. The purified RNA was stored at −80°C until use. RT–qPCR was performed on the QuantStudio 5 system (Thermo Fisher Scientific) using Luna Universal One-Step RT–qPCR Kit (NEB). All reactions were performed in a total volume of 15 $\mu$l consisting of 1X Luna Universal One-Step Reaction Mix, 0.75 $\mu$l of Luna WarmStart RT Enzyme Mix, 0.1 $\mu$M of each of the forward and reverse primers, and 2 $\mu$l of template RNA. Previously reported forward (5′-GCGTTCTTCGGAATGTCG-3′) and reverse (5′-TTGGATCTTTGTCATCCAATTTG-3′) primer set targeting the *N*-gene was used (Chan et al, 2020). The cycling conditions consisted of 15 min at 55°C and an initial denaturation of 2 min at 95°C followed by 40 cycles of 15 s at 95°C and 60 s at 60°C. The specificity of the resulting amplicons was determined using the melting curve. RT-LAMP was performed using lyophilized RT-LAMP reagent mix shipped from Vienna, Austria. The mix was reconstituted on-site, and extracted RNA, as well as heat-inactivated samples, was tested by incubating RT-LAMP reactions at 63°C for 35 min in a real-time thermocycler, with fluorescence readings taken in the FAM/SYBR spectrum every 60 s. The colorimetric results were scored by the researcher immediately after incubation using the colorimetry.net web app.

## Data Availability

The original contributions presented in the study are included in the Article, Supplementary Material, and Source Data file. Further inquiries can be directed to the corresponding authors.

## Supplementary Information

## Acknowledgements

This work would not have been possible without the enthusiastic support of the IMBA, IMP, and WACCBIP research institutes, as well as the many volunteers and partners of the VCDI, who came together to help and collaborate under the exceptional circumstances of the COVID-19 pandemic. We thank the Pauli, Brennecke, and Penninger groups for sharing laboratory space and reagents. We thank Nathan Tanner (NEB) for valuable discussions, sharing important information on LAMP technology, and feedback on the article, Stuart Le Grice and Jennifer Miller (NIH/NCI) for helpful advice and reagents for the expression of HIV-RTs, and Jennifer C Molloy (University of Cambridge) for useful discussion. We thank TReND in Africa for initiating the collaboration between our laboratories in Vienna and Ghana, and

Jochen Wittbrodt, Ute Volbehr, Kevin Urbansky, and Eva Hasel for administering the VW funding. We also thank Kim and Anna Nasmyth for their generous support of the project via the "Mila Charitable Organisation". We are grateful to the COVID Testing Scaleup Slack channel for openly sharing and exchanging information. M Matl and MJ Kellner were supported by the Vienna Science and Technology Fund (WWTF) through project COV20-031. This project received funding from the VW foundation (grant # 9A027) given to TReND in Africa, as well as funding from the "Mila Charitable Organisation". The IMP receives generous institutional funding from Boehringer Ingelheim and the Austrian Research Promotion Agency (Headquarter grant FFG-852936), and IMBA is generously supported by the Austrian Academy of Sciences. Research in the laboratory of A Pauli was supported by the Austrian Science Fund (START Projekt Y 1031-B28, SFB "RNA-Deco" F 80) and by the European Research Council (CoG 101044495/GaMe). Research in the laboratory of J Brennecke was supported by the European Research Council (ERC2015-CoG - 682181). F Ansah and GA Awandare were supported by a Science for Africa Foundation (SFA)/Wellcome Developing Excellence in Leadership Training and Science (DELTAS) in Africa grant (DEL-22-014: GA Awandare). Work in the L Menéndez-Arias laboratory was supported by grant PID2022-136725OB-I00/AEI/10.13039/501100011033 of the Spanish Ministry of Science, Innovation and Universities, and an institutional grant of Fundación Ramón Areces. The laboratory of JM Penninger at IMBA received funding from the Austrian Academy of Sciences, the Medical University of Vienna, the Swedish Research Council (2018-05766), the T. von Zastrow foundation, and the Innovative Medicines Initiative 2 Joint Undertaking under grant agreement No. 101005026. This Joint Undertaking receives support from the European Union's Horizon 2020 research and innovation programme and EFPIA.

## Author Contributions

M Matl: conceptualization, data curation, formal analysis, validation, investigation, methodology, and writing—original draft, review, and editing.

MJ Kellner: conceptualization, data curation, formal analysis, validation, investigation, methodology, and writing—original draft, review, and editing.

F Ansah: conceptualization, data curation, formal analysis, validation, investigation, methodology, and writing—original draft, review, and editing.

I Grishkovskaya: investigation and methodology.

D Handler: investigation and methodology.

R Heinen: investigation and methodology.

B Bauer: methodology.

L Menéndez-Arias: resources and methodology.

TO Auer: resources and funding acquisition.

LL Prieto-Godino: resources and funding acquisition.

JM Penninger: resources and funding acquisition.

VC-19 Vienna COVID-19 Detection Initiative: resources and methodology.

GA Awandare: resources and supervision.

J Brennecke: conceptualization, resources, supervision, funding acquisition, project administration, and writing—review and editing.

A Pauli: conceptualization, resources, supervision, funding acquisition, project administration, and writing—review and editing.

## Conflict of Interest Statement

The authors declare that they have no conflict of interest.

# Members of the Vienna COVID-19 Detection Initiative (VCDI)

Stefan Ameres, Institute of Molecular Biotechnology of the Austrian Academy of Sciences (IMBA), Vienna Biocenter (VBC), Vienna, Austria; Benedikt Bauer, Research Institute of Molecular Pathology (IMP), Vienna Biocenter (VBC), Vienna, Austria; Nikolaus Beer, Research Institute of Molecular Pathology (IMP), Vienna Biocenter (VBC), Vienna, Austria, Institute of Molecular Biotechnology of the Austrian Academy of Sciences (IMBA), Vienna Biocenter (VBC), Vienna, Austria, and Gregor Mendel Institute (GMI), Austrian Academy of Sciences, Vienna Biocenter (VBC), Vienna, Austria; Katharina Bergauer, Research Institute of Molecular Pathology (IMP), Vienna Biocenter (VBC), Vienna, Austria; Wolfgang Binder, Max Perutz Labs, Medical University of Vienna, Vienna Biocenter (VBC), Vienna, Austria; Claudia Blaukopf, Institute of Molecular Biotechnology of the Austrian Academy of Sciences (IMBA), Vienna Biocenter (VBC), Vienna, Austria; Boril Bochev, Research Institute of Molecular Pathology (IMP), Vienna Biocenter (VBC), Vienna, Austria, Institute of Molecular Biotechnology of the Austrian Academy of Sciences (IMBA), Vienna Biocenter (VBC), Vienna, Austria, and Gregor Mendel Institute (GMI), Austrian Academy of Sciences, Vienna Biocenter (VBC), Vienna, Austria; Julius Brennecke, Institute of Molecular Biotechnology of the Austrian Academy of Sciences (IMBA), Vienna Biocenter (VBC), Vienna, Austria; Selina Brinnich, Vienna Biocenter Core Facilities GmbH (VBCF), Vienna, Austria; Aleksandra Bundalo, Research Institute of Molecular Pathology (IMP), Vienna Biocenter (VBC), Vienna, Austria; Meinrad Busslinger, Research Institute of Molecular Pathology (IMP), Vienna Biocenter (VBC), Vienna, Austria; Aleksandr Bykov, Research Institute of Molecular Pathology (IMP), Vienna Biocenter (VBC), Vienna, Austria; Tim Clausen, Research Institute of Molecular Pathology (IMP), Vienna Biocenter (VBC), Vienna, Austria, and Medical University of Vienna, Vienna Biocenter (VBC), Vienna, Austria; Luisa Cochella, Research Institute of Molecular Pathology (IMP), Vienna Biocenter (VBC), Vienna, Austria; Geert de Vries, Institute of Molecular Biotechnology of the Austrian Academy of Sciences (IMBA), Vienna Biocenter (VBC), Vienna, Austria; Marcus Dekens, Research Institute of Molecular Pathology (IMP), Vienna Biocenter (VBC), Vienna, Austria; David Drechsel, Research Institute of Molecular Pathology (IMP), Vienna Biocenter (VBC), Vienna, Austria; Zuzana Dzupinkova, Research Institute of Molecular Pathology (IMP), Vienna Biocenter (VBC), Vienna, Austria, Institute of Molecular Biotechnology of the Austrian Academy of Sciences (IMBA), Vienna Biocenter (VBC), Vienna, Austria, and Gregor Mendel Institute (GMI), Austrian Academy of Sciences, Vienna Biocenter (VBC), Vienna, Austria; Michaela Eckmann-Mader, Vienna Biocenter Core Facilities GmbH (VBCF), Vienna, Austria; Ulrich Elling, Institute of Molecular Biotechnology of the Austrian Academy of Sciences (IMBA), Vienna Biocenter (VBC), Vienna, Austria; Michaela Fellner, Research Institute of Molecular Pathology (IMP), Vienna Biocenter (VBC), Vienna, Austria; Thomas Fellner, Vienna Biocenter Core Facilities GmbH (VBCF), Vienna, Austria; Laura Fin, Research Institute of Molecular Pathology (IMP), Vienna Biocenter (VBC), Vienna, Austria; Bianca Valeria Gapp, Institute of Molecular Biotechnology of the Austrian Academy of Sciences (IMBA), Vienna Biocenter (VBC),

Vienna, Austria; Gerlinde Grabmann, Vienna Biocenter Core Facilities GmbH (VBCF), Vienna, Austria; Irina Grishkovskaya, Research Institute of Molecular Pathology (IMP), Vienna Biocenter (VBC), Vienna, Austria; Astrid Hagelkruys, Institute of Molecular Biotechnology of the Austrian Academy of Sciences (IMBA), Vienna Biocenter (VBC), Vienna, Austria; Bence Hajdusits, Research Institute of Molecular Pathology (IMP), Vienna Biocenter (VBC), Vienna, Austria; David Haselbach, Research Institute of Molecular Pathology (IMP), Vienna Biocenter (VBC), Vienna, Austria; Robert Heinen, Research Institute of Molecular Pathology (IMP), Vienna Biocenter (VBC), Vienna, Austria, Institute of Molecular Biotechnology of the Austrian Academy of Sciences (IMBA), Vienna Biocenter (VBC), Vienna, Austria, and Gregor Mendel Institute (GMI), Austrian Academy of Sciences, Vienna Biocenter (VBC), Vienna, Austria; Louisa Hill, Research Institute of Molecular Pathology (IMP), Vienna Biocenter (VBC), Vienna, Austria; David Hoffmann, Institute of Molecular Biotechnology of the Austrian Academy of Sciences (IMBA), Vienna Biocenter (VBC), Vienna, Austria; Stefanie Horer, Research Institute of Molecular Pathology (IMP), Vienna Biocenter (VBC), Vienna, Austria; Harald Isemann, Research Institute of Molecular Pathology (IMP), Vienna Biocenter (VBC), Vienna, Austria; Robert Kalis, Research Institute of Molecular Pathology (IMP), Vienna Biocenter (VBC), Vienna, Austria; Max Kellner, Research Institute of Molecular Pathology (IMP), Vienna Biocenter (VBC), Vienna, Austria, and Institute of Molecular Biotechnology of the Austrian Academy of Sciences (IMBA), Vienna Biocenter (VBC), Vienna, Austria; Juliane Kley, Research Institute of Molecular Pathology (IMP), Vienna Biocenter (VBC), Vienna, Austria; Thomas Köcher, Vienna Biocenter Core Facilities GmbH (VBCF), Vienna, Austria; Alwin Köhler, Max Perutz Labs, Medical University of Vienna, Vienna Biocenter (VBC), Vienna, Austria; Darja Kordic, Research Institute of Molecular Pathology (IMP), Vienna Biocenter (VBC), Vienna, Austria; Christian Krauditsch, Institute of Molecular Biotechnology of the Austrian Academy of Sciences (IMBA), Vienna Biocenter (VBC), Vienna, Austria; Sabina Kula, Research Institute of Molecular Pathology (IMP), Vienna Biocenter (VBC), Vienna, Austria, Institute of Molecular Biotechnology of the Austrian Academy of Sciences (IMBA), Vienna Biocenter (VBC), Vienna, Austria, and Gregor Mendel Institute (GMI), Austrian Academy of Sciences, Vienna Biocenter (VBC), Vienna, Austria; Richard Latham, Research Institute of Molecular Pathology (IMP), Vienna Biocenter (VBC), Vienna, Austria; Marie-Christin Leitner, Institute of Molecular Biotechnology of the Austrian Academy of Sciences (IMBA), Vienna Biocenter (VBC), Vienna, Austria; Thomas Leonard, Max Perutz Labs, Medical University of Vienna, Vienna Biocenter (VBC), Vienna, Austria; Dominik Lindenhofer, Institute of Molecular Biotechnology of the Austrian Academy of Sciences (IMBA), Vienna Biocenter (VBC), Vienna, Austria; Raphael Arthur Manzenreither, Institute of Molecular Biotechnology of the Austrian Academy of Sciences (IMBA), Vienna Biocenter (VBC), Vienna, Austria; Karl Mechtler, Research Institute of Molecular Pathology (IMP), Vienna Biocenter (VBC), Vienna, Austria; Anton Meinhart, Research Institute of Molecular Pathology (IMP), Vienna Biocenter (VBC), Vienna, Austria; Stefan Mereiter, Institute of Molecular Biotechnology of the Austrian Academy of Sciences (IMBA), Vienna Biocenter (VBC), Vienna, Austria; Thomas Micheler, Vienna Biocenter Core Facilities GmbH (VBCF), Vienna, Austria; Paul Moeseneder, Institute of Molecular Biotechnology of the Austrian Academy of Sciences (IMBA), Vienna Biocenter (VBC), Vienna, Austria; Tobias Neumann, Research Institute of Molecular Pathology (IMP), Vienna Biocenter (VBC), Vienna, Austria; Simon Nimpf, Research Institute of Molecular Pathology (IMP), Vienna Biocenter (VBC), Vienna, Austria; Magnus Nordborg, Gregor Mendel Institute (GMI), Austrian Academy of Sciences, Vienna Biocenter (VBC), Vienna, Austria; Egon Ogris, Max Perutz Labs, Medical University of Vienna, Vienna Biocenter (VBC), Vienna, Austria; Michaela Pagani, Research Institute of Molecular Pathology (IMP), Vienna Biocenter (VBC), Vienna, Austria; Andrea Pauli, Research Institute of Molecular Pathology (IMP), Vienna Biocenter (VBC), Vienna, Austria; JanMichael Peters, Research Institute of Molecular Pathology (IMP), Vienna Biocenter (VBC), Vienna, Austria, and Medical University of Vienna, Vienna Biocenter (VBC), Vienna, Austria; Petra Pjevac, Centre for Microbiology and Environmental Systems Science, University of Vienna, Vienna, Austria, and Joint Microbiome Facility of the University of Vienna and Medical University of Vienna, Vienna, Austria; Clemens Plaschka, Research Institute of Molecular Pathology (IMP), Vienna Biocenter (VBC), Vienna, Austria; Martina Rath, Research Institute of Molecular Pathology (IMP), Vienna Biocenter (VBC), Vienna, Austria; Daniel Reumann, Institute of Molecular Biotechnology of the Austrian Academy of Sciences (IMBA), Vienna Biocenter (VBC), Vienna, Austria; Sarah Rieser, Research Institute of Molecular Pathology (IMP), Vienna Biocenter (VBC), Vienna, Austria; Marianne Rocha-Hasler, Centre for Microbiology and Environmental Systems Science, University of Vienna, Vienna, Austria; Alan Rodriguez, Research Institute of Molecular Pathology (IMP), Vienna Biocenter (VBC), Vienna, Austria, and Institute of Molecular Biotechnology of the Austrian Academy of Sciences (IMBA), Vienna Biocenter (VBC), Vienna, Austria; James Julian Ross, Institute of Molecular Biotechnology of the Austrian Academy of Sciences (IMBA), Vienna Biocenter (VBC), Vienna, Austria; Harald Scheuch, Research Institute of Molecular Pathology (IMP), Vienna Biocenter (VBC), Vienna, Austria, Institute of Molecular Biotechnology of the Austrian Academy of Sciences (IMBA), Vienna Biocenter (VBC), Vienna, Austria, and Gregor Mendel Institute (GMI), Austrian Academy of Sciences, Vienna Biocenter (VBC), Vienna, Austria; Karina Schindler, Research Institute of Molecular Pathology (IMP), Vienna Biocenter (VBC), Vienna, Austria; Clara Schmidt, Institute of Molecular Biotechnology of the Austrian Academy of Sciences (IMBA), Vienna Biocenter (VBC), Vienna, Austria; Hannes Schmidt, Centre for Microbiology and Environmental Systems Science, University of Vienna, Vienna, Austria; Jakob Schnabl, Institute of Molecular Biotechnology of the Austrian Academy of Sciences (IMBA), Vienna Biocenter (VBC), Vienna, Austria; Stefan Schüchner, Max Perutz Labs, Medical University of Vienna, Vienna Biocenter (VBC), Vienna, Austria; Tanja Schwickert, Research Institute of Molecular Pathology (IMP), Vienna Biocenter (VBC), Vienna, Austria; Andreas Sommer, Vienna Biocenter Core Facilities GmbH (VBCF), Vienna, Austria; Johannes Stadlmann, Institute of Biochemistry, University of Natural Resources and Life Sciences (BOKU), Vienna, Austria; Alexander Stark, Research Institute of Molecular Pathology (IMP), Vienna Biocenter (VBC), Vienna, Austria, and Medical University of Vienna, Vienna Biocenter (VBC), Vienna, Austria; Peter Steinlein, Research Institute of Molecular Pathology (IMP), Vienna Biocenter (VBC), Vienna, Austria, Institute of Molecular Biotechnology of the

Austrian Academy of Sciences (IMBA), Vienna Biocenter (VBC), Vienna, Austria, and Gregor Mendel Institute (GMI), Austrian Academy of Sciences, Vienna Biocenter (VBC), Vienna, Austria; Simon Strobl, Vienna Biocenter Core Facilities GmbH (VBCF), Vienna, Austria; Qiong Sun, Research Institute of Molecular Pathology (IMP), Vienna Biocenter (VBC), Vienna, Austria; Wen Tang, Research Institute of Molecular Pathology (IMP), Vienna Biocenter (VBC), Vienna, Austria; Linda Trübestein, Max Perutz Labs, Medical University of Vienna, Vienna Biocenter (VBC), Vienna, Austria; Christian Umkehrer, Research Institute of Molecular Pathology (IMP), Vienna Biocenter (VBC), Vienna, Austria; Sandor Urmosi-Incze, Vienna Biocenter Core Facilities GmbH (VBCF), Vienna, Austria; Kristina Uzunova, Research Institute of Molecular Pathology (IMP), Vienna Biocenter (VBC), Vienna, Austria, Institute of Molecular Biotechnology of the Austrian Academy of Sciences (IMBA), Vienna Biocenter (VBC), Vienna, Austria, and Gregor Mendel Institute (GMI), Austrian Academy of Sciences, Vienna Biocenter (VBC), Vienna, Austria; Gijs Versteeg, Department of Microbiology, Immunobiology, and Genetics, Max Perutz Labs, University of Vienna, Vienna Biocenter (VBC), Dr. Bohr-Gasse 9, 1,030 Vienna, Austria; Alexander Vogt, Vienna Biocenter Core Facilities GmbH (VBCF), Vienna, Austria; Vivien Vogt, Research Institute of Molecular Pathology (IMP), Vienna Biocenter (VBC), Vienna, Austria; Michael Wagner, Centre for Microbiology and Environmental Systems Science, University of Vienna, Vienna, Austria, and Joint Microbiome Facility of the University of Vienna and Medical University of Vienna, Vienna, Austria; Martina Weissenboeck, Research Institute of Molecular Pathology (IMP), Vienna Biocenter (VBC), Vienna, Austria; Barbara Werner, Vienna Biocenter Core Facilities GmbH (VBCF), Vienna, Austria; Ramesh Yelagandula, Institute of Molecular Biotechnology of the Austrian Academy of Sciences (IMBA), Vienna Biocenter (VBC), Vienna, Austria; Johannes Zuber, Research Institute of Molecular Pathology (IMP), Vienna Biocenter (VBC), Vienna, Austria, and Medical University of Vienna, Vienna Biocenter (VBC), Vienna, Austria.

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
