## [Reviewer comments · Life Science Alliance]

Life Science Alliance

A lyophilized open-source RT-LAMP assay for molecular diagnostics in resource-limited settings

Martin Matl, Max Kellner, Felix Ansah, Irina Grishkovskaya, Dominik Handler, Robert Heinen, Benedikt Bauer, Luis Menendez-Arias, Thomas Auer, Lucia Prieto-Godino, Josef Penninger, Vienna Vienna Covid-19 Detection Initiative, Gordon Awandare, Julius Brennecke, and Andrea Pauli

DOI: <https://doi.org/10.26508/lsa.202403167>

Corresponding author(s): Andrea Pauli, Research Institute of Molecular Pathology; Julius Brennecke, IMBA; and Gordon Awandare, University of Ghana

Review Timeline:

Submission Date:	2024-12-09
Editorial Decision:	2025-01-21
Revision Received:	2025-05-31
Editorial Decision:	2025-06-20
Revision Received:	2025-06-28
Accepted:	2025-07-01

Scientific Editor: Tim Fessenden

Transaction Report:

January 21, 2025

Re: Life Science Alliance manuscript #LSA-2024-03167-T

Dr. Andrea Pauli
IMP, Vienna, Austria
Vienna Biocenter (VBC)
Campus-Vienna-Biocenter 1
Vienna, Vienna 1030
Austria

Dear Dr. Pauli,

Thank you for submitting your manuscript entitled "A lyophilized open-source RT-LAMP assay for molecular diagnostics in resource-limited settings" to Life Science Alliance. The manuscript was assessed by expert reviewers, whose comments are appended to this letter. We invite you to submit a revised manuscript addressing the Reviewer comments.

Thank you for this interesting contribution to Life Science Alliance. We are looking forward to receiving your revised manuscript.

Sincerely,

B. MANUSCRIPT ORGANIZATION AND FORMATTING:

Reviewer #1 (Comments to the Authors (Required)):

This manuscript describes an effort to create an open-source version of an RT-LAMP assay workflow for detection of SARS-CoV-2 from swab samples, to include non-proprietary enzymes, buffers, quick-extraction reagents, and lyophilization mixtures, with the intent of improving accessibility to diagnostic tests in low and middle income countries. There are a variety of ways I could see this working: a non-profit organization could produce these reagents at large scale and distribute at-cost and royalty-free to partners in low-resource countries, or (since the "recipes" would all be freely published), manufacturing of the reagents can be done in a distributed fashion directly in low-and-middle income countries.

This all sounds laudable. One of the "value adds" of having commercial producers of reagents like NEB, Promega, etc is having robust quality control (QC). In the developed world, we require manufacturers of diagnostic tests to have robust QC, which is captured as various programs such as good manufacturing & good laboratory practices, various ISO certifications, etc. Having such programs costs money, although in theory it's money well-spent in the interest of having reliable and reproducible results from diagnostic tests, and I don't believe low-and-middle income countries should be expected to make do with less robust diagnostic tests. It would seem to me that it would be easier and more cost effective to set up a QC program for a single central manufacturer that distributes reagents, vs trying to have many smaller in-country QC programs. And so I think the model demonstrated in this manuscript of having the lab in Vienna produce reagents and then test the reagents at two sites (in Vienna and Ghana) is probably the more workable model. An interesting follow-on to this work however would be an analysis (including a technoeconomic analysis) of various models for use of the products of this research, including what parts of the supply chain are vulnerable to disruptions, and whether there are any roadblocks to a more distributed production model (e.g. availability of oligo primers?)

Onto technical matters: The overall work is quite comprehensive in scope. The level of characterization is ok for academic literature, and I don't have major concerns with the results, although it falls short of what would be expected for clinical use of a diagnostic product (at least in the USA). Much of the ground they discuss (especially on quick extraction) has been covered elsewhere although I'm not sure I've seen betaine added into the mix for quick extractions.

Experiments described in Fig 1A/1B on evaluating enzymes with 4 replicates are not sufficient to really say with any confidence that these mixtures are giving different performance. If they went through the exercise of performing a probit fit to their data and calculating LOD50 or LOD95 and confidence intervals they would find very wide and likely overlapping confidence intervals. So for example in Fig 1B, where MMuLV and AMV RT bounce up and down instead of having a monotonic trend, this is an artifact of only doing 4 replicates per concentration, and the inherently stochastic nature of these experiments, versus any "real" behavior of the enzymes. Thus, statements in text like "Strikingly, our in-house purified HIV-1 RT showed equal speed and sensitivity as WarmStart® RTx" should be toned down. I think the most they can say is that within the statistical power of the experiments, the in-house enzymes give similar performance to the commercial enzymes.

I'd also caution that when comparing enzymes, results (in my humble experience) are not translatable from one assay to the next. So for example, I've had the experience where Bst 3 absolutely crushes with one primer set, it may generate a lot of garbage with another, and sometimes Bst 2 gives the fastest results, and sometimes there is no discernible difference between the engineered enzymes and the original. So the results reported herein should not be viewed as generalizable across assays other than the ones tested here.

I've also observed supplier-to-supplier differences for individual enzymes, for example AMV RT from a certain supplier consistently outperformed AMV RT from other suppliers (as well as WarmStart RTx), on a unit-for-unit basis. It suggests to me that difference in production methods can lead to substantially different results for what is nominally the same enzyme. So the result here is not so much that HIV RT is better than AMV RT is better than MMuLV RT, but rather HIV RT "as we produced it" outperforms the others "as we produced them". I'm a little surprised the HIV RT outperformed the AMV RT, but did I catch in there that the HIV RT is a thermostable variant? In many tests I have found AMV RT always outperforms MMuLV RT (including thermally stable, RNase H+ or - engineered versions), but I have never tried HIV RT.

If I understood correctly, the BeadLamp experiments use a commercial product, RNAClean XP beads, from Beckman-Coulter. Is

there an open-source version of these? I wonder if here the authors are up against the limits of what can be done with open-source methods!

The experiments on lyophilization are not that ambitious in demonstrating stability, in that they test out to a month at room temperature. Temperatures in shipping & storage (especially in developing world) can get much hotter, and they will likely want to consider longer shelf lives. As a preliminary study this is fine, but one thing to consider would be accelerated aging studies, e.g. store some aliquots at 30, 40, 50, 60 C and see if they can deduce a time vs temperature relationship to deactivation that would enable them to extrapolate out to longer times. However that is not necessary for the present work.

There is probably still a significant difference between medium-scale lab production as described here (i.e. protein expression at scale of 4 L) and true industrial production. The lyophilization is likely a procedure where industrial-scale equipment will do a better job than what they can accomplish in the lab. For example, being able to backfill the tubes with dry nitrogen and stopper directly within the chamber would be an advantage vs flushing with possibly humid atmosphere. A more industrial operation would also be able to vacuum-seal the tubes in moisture proof pouches.

The color transformations for the HNB color change are interesting, in that I hadn't seen that before. I assume this is purely to aid the human eye in visualizing the difference, vs the original RGB image, but to me it seems this is 'after-the-fact', i.e. there is no new information being generated by the image processing, but rather a magnification or rescaling and false-coloring of the difference in hue, but this is all accomplished by image processing or machine vision that is independent of human perception.

Reviewer #2 (Comments to the Authors (Required)):

This paper by the Brennecke and Pauli labs describes the development and validation of a sensitive, robust RT-LAMP-based assay for the detection of SARS-CoV-2 RNA with good to excellent sensitivity. Unlike the plethora of RT-LAMP papers that have emerged in the aftermath of the pandemic, this paper merits publication for several compelling reasons:

- The paper addresses the challenge of creating a field-compatible RT-LAMP assay using only non-proprietary, low-cost ingredients.
- It provides a sample lysis/processing solution for direct testing that appears to be equivalent to commercial (proprietary) formulations.
- It describes procedures for direct testing of swab and gargle samples with good sensitivity.
- It details an enzyme mixture for RT-LAMP that can be self-produced and demonstrates sensitivity comparable to commercial formulations.
- It incorporates measures to protect against contamination, a critical consideration in non-laboratory settings.
- The procedures developed include lyophilized formulations for scenarios where a continuous cold chain for enzyme/reagent delivery cannot be guaranteed.
- The work is executed to a high standard.
- The work emphasizes reproducibility and applicability, including a webpage with detailed supplementary information.
- The study constitutes a valuable resource for future pandemics.

The 'downside' of the method is that it has not yet been applied to a large set of samples in a SARS-CoV-2 pandemic setting (and it is likely never will be - which, in fact, is good news!). However, new pandemics may arise, and this work will undoubtedly prove very useful in such scenarios in the future.

What I find missing is one or two sections discussing critical points to prepare future users (which may be molecular biologists with little/zero experience in diagnostics) who might need to rapidly adapt the method to new scenarios:

- Primer design is crucial-good LAMP primers make all the difference (and designing them is notoriously challenging).
- The fact that a sensitivity of CT=31 may suffice for one application but not for others, such as acute testing of symptomatic patients versus surveillance testing in a low-incidence setting.
- Different viruses/pathogens may have varying requirements for sensitivity, as viral titers could generally be lower.
- Including a short perspective on potential future improvements would further strengthen the paper.

Altogether, the work can be published with minimal modifications.

Reviewer #3 (Comments to the Authors (Required)):

This study addresses significant challenges in achieving equitable access to molecular diagnostics. Firstly, the development of open-source, cost-effective assays for large-scale nucleic acid detection and secondly the creation of lyophilized reagent formulations for cold chain-independent storage and shipment.

The researchers have developed a sensitive, rapid and accurate RT-LAMP assay for virus detection (modelled on SARS-CoV-2), demonstrating non-proprietary enzymatic components that can be produced and lyophilized in standard laboratory settings - highly suitable for resource-limited environments. Data was clearly presented and strongly supportive.

The study contributes several advances to the field namely open-access enzymes and protocols, available on an open-access platform. These protocols empower laboratories to produce reagents independently, lowering reliance on commercial suppliers. The reagents and protocols were successfully tested on other respiratory viruses, highlighting their potential for pathogen surveillance.

Independent benchmarking studies in Vienna and Ghana confirmed the assay's high sensitivity and specificity using both contrived and clinical samples, demonstrating its robustness across varying conditions.

This work helps to pave the way for equitable access to diagnostics, strengthening global capabilities for pathogen surveillance and epidemic management.

We thank the editorial team for considering our manuscript for publication in Life Science Alliance and we thank the reviewers for their valuable feedback, which has helped us to further improve our manuscript. Based on the feedback provided, we have made the following modifications in addition to some minor adjustments:

- We have newly added experiments showing the titration of *Bst* LF, HIV-1 RT and BMTU UDG in LAMP reactions for optimization of reaction performance and quality control (new **Supplementary Figures 2F-H**).
- We have toned down statements regarding the performance of open source enzymes compared to commercial enzymes in accordance with the statistical strength of the experiments (text of the Results section regarding Figures 1A,B).
- We have expanded our discussion to comment on additional challenges in implementing RT-LAMP in low-resource settings, and to include a link to rtlamp.org, where we host our practical knowledge of RT-LAMP including protocols, how-to guides and videos aimed for people without prior experience with diagnostics or LAMP.

Our point-by-point responses to the comments are provided below.

Reviewer #1 (Comments to the Authors (Required)):

This manuscript describes an effort to create an open-source version of an RT-LAMP assay workflow for detection of SARS-CoV-2 from swab samples, to include non-proprietary enzymes, buffers, quick-extraction reagents, and lyophilization mixtures, with the intent of improving accessibility to diagnostic tests in low and middle income countries. There are a variety of ways I could see this working: a non-profit organization could produce these reagents at large scale and distribute at-cost and royalty-free to partners in low-resource countries, or (since the "recipes" would all be freely published), manufacturing of the reagents can be done in a distributed fashion directly in low-and-middle income countries.

We thank the reviewer for the appreciation of our work. We fully agree with the possible modes of distribution of this open-source molecular diagnostic assay that the reviewer has outlined here. We would like to add that the protocols and additional information regarding the procedures for successfully implementing RT-LAMP are available on our website rtlamp.org.

This all sounds laudable. One of the "value adds" of having commercial producers of reagents like NEB, Promega, etc is having robust quality control (QC). In the developed

world, we require manufacturers of diagnostic tests to have robust QC, which is captured as various programs such as good manufacturing & good laboratory practices, various ISO certifications, etc. Having such programs costs money, although in theory it's money well-spent in the interest of having reliable and reproducible results from diagnostic tests, and I don't believe low-and-middle income countries should be expected to make do with less robust diagnostic tests. It would seem to me that it would be easier and more cost effective to set up a QC program for a single central manufacturer that distributes reagents, vs trying to have many smaller in-country QC programs. And so I think the model demonstrated in this manuscript of having the lab in Vienna produce reagents and then test the reagents at two sites (in Vienna and Ghana) is probably the more workable model. An interesting follow-on to this work however would be an analysis (including a technoeconomic analysis) of various models for use of the products of this research, including what parts of the supply chain are vulnerable to disruptions, and whether there are any roadblocks to a more distributed production model (e.g. availability of oligo primers?)

We thank the reviewer for highlighting the critical role of quality control frameworks in ensuring robust diagnostics. We fully agree that quality control is essential, and we have added a section acknowledging the importance of it to the Discussion.

While a comprehensive techno-economic analysis and exploration of supply chain vulnerabilities would indeed be valuable, such analyses are beyond the current scope of this manuscript.

Onto technical matters: The overall work is quite comprehensive in scope. The level of characterization is ok for academic literature, and I don't have major concerns with the results, although it falls short of what would be expected for clinical use of a diagnostic product (at least in the USA). Much of the ground they discuss (especially on quick extraction) has been covered elsewhere although I'm not sure I've seen betaine added into the mix for quick extractions.

We thank the reviewer for their comment. In our hands, betaine, although a common additive to both LAMP and PCR, has turned out to be a useful and novel addition to a quick extraction solution since it increased assay robustness.

Experiments described in Fig 1A/1B on evaluating enzymes with 4 replicates are not sufficient to really say with any confidence that these mixtures are giving different performance. If they went through the exercise of performing a probit fit to their data and calculating LOD50 or LOD95 and confidence intervals they would find very wide and likely

overlapping confidence intervals. So for example in Fig 1B, where MMuLV and AMV RT bounce up and down instead of having a monotonic trend, this is an artifact of only doing 4 replicates per concentration, and the inherently stochastic nature of these experiments, versus any "real" behavior of the enzymes. Thus, statements in text like "Strikingly, our in-house purified HIV-1 RT showed equal speed and sensitivity as WarmStart® RTx" should be toned down. I think the most they can say is that within the statistical power of the experiments, the in-house enzymes give similar performance to the commercial enzymes.

Thank you for highlighting this point. We agree that with an n=4 we cannot confidently state that one enzyme outperforms another one in the given conditions with the stochasticity that is seen in LAMP reaction outcomes near the limit of detection. Therefore, we've decided to tone down the statements of Figure 1A and Figure 1B.

I'd also caution that when comparing enzymes, results (in my humble experience) are not translatable from one assay to the next. So for example, I've had the experience where Bst 3 absolutely crushes with one primer set, it may generate a lot of garbage with another, and sometimes Bst 2 gives the fastest results, and sometimes there is no discernible difference between the engineered enzymes and the original. So the results reported herein should not be viewed as generalizable across assays other than the ones tested here.

We agree that enzyme performance can be somewhat assay- and primer-dependent, and that results obtained in one context may not generalize to others. While we did not observe substantial variability in our SARS-CoV-2 assays, we recognize the importance of this caveat. Accordingly, we clarified in our manuscript that our enzyme comparison is specific to the SARS-CoV-2 assays tested here.

I've also observed supplier-to-supplier differences for individual enzymes, for example AMV RT from a certain supplier consistently outperformed AMV RT from other suppliers (as well as WarmStart RTx), on a unit-for-unit basis. It suggests to me that difference in production methods can lead to substantially different results for what is nominally the same enzyme. So the result here is not so much that HIV RT is better than AMV RT is better than MMuLV RT, but rather HIV RT "as we produced it" outperforms the others "as we produced them". I'm a little surprised the HIV RT outperformed the AMV RT, but did I catch in there that the HIV RT is a thermostable variant? In many tests I have found AMV RT always outperforms MMuLV RT (including thermally stable, RNase H+ or - engineered versions), but I have never tried HIV RT.

Supplier differences are indeed a factor that needs to be considered when comparing enzymes and their activity. While we cannot control for batch differences per se, we did

attempt to make our results as standardized as possible by comparing enzymes from one supplier against our enzymes. For the DNA polymerases test, commercial *Bst* DNA Polymerase Large Fragment, *Bst* 2.0 DNA Polymerase, *Bst* 2.0 WarmStart® DNA Polymerase and *Bst* 3.0 DNA Polymerase were all from New England Biolabs and were used at the concentrations recommended by the manufacturer. For the reverse transcriptase test, the AMV Reverse Transcriptase, MMuLV Reverse Transcriptase and RTx Reverse Transcriptase were also purchased from New England Biolabs and used at recommended concentrations. We think that this approach is reasonable and fair, although we do admit that the commercial enzymes could perform better or worse at different concentrations.

HIV-1 RT is in our experience a reliable enzyme at the elevated temperatures used for RT-LAMP (60-65°C; AMV RT does not perform well beyond 50-55°C in our hands). While thermostable variants of HIV-1 RT do exist, in the current work we present here we used a wild type non-engineered HIV-1 RT from the BH-10 strain that should therefore not be classified as “thermostable”. We’ve removed this word from the manuscript and apologize for the confusion. It was merely meant to convey that HIV-1 RT can facilitate cDNA synthesis at elevated temperatures.

If I understood correctly, the BeadLamp experiments use a commercial product, RNAClean XP beads, from Beckman-Coulter. Is there an open-source version of these? I wonder if here the authors are up against the limits of what can be done with open-source methods!

For the experiments presented in the manuscript, we actually utilized silica-coated magnetic beads prepared in-house using protocols published by Oberacker et al., 2019, whom we cite in the Results section where we talk about using in-house and open access beads for this purpose. We realized after checking now that we erroneously listed Beckman-Coulter beads in the Method section for bead.LAMP. This was an error on our side since we had originally developed the protocol using commercial beads. We’ve modified the Methods section to correct this mistake and state that in-house beads were used.

The experiments on lyophilization are not that ambitious in demonstrating stability, in that they test out to a month at room temperature. Temperatures in shipping & storage (especially in developing world) can get much hotter, and they will likely want to consider longer shelf lives. As a preliminary study this is fine, but one thing to consider would be accelerated aging studies, e.g. store some aliquots at 30, 40, 50, 60 C and see if they can deduce a time vs temperature relationship to deactivation that would enable them to extrapolate out to longer times. However that is not necessary for the present work.

We thank the reviewer for their suggestion and agree with the assessment that an accelerated aging study would be beneficial to showcase the stability of lyophilized open source RT-LAMP assays. While our tests after storage at 37°C for 40 days offer already some insights in this regard, we appreciate the reviewer's understanding that performing more advanced tests on our enzyme mixes would take a considerable amount of time that goes beyond the scope of this manuscript.

There is probably still a significant difference between medium-scale lab production as described here (i.e. protein expression at scale of 4 L) and true industrial production. The lyophilization is likely a procedure where industrial-scale equipment will do a better job than what they can accomplish in the lab. For example, being able to backfill the tubes with dry nitrogen and stopper directly within the chamber would be an advantage vs flushing with possibly humid atmosphere. A more industrial operation would also be able to vacuum-seal the tubes in moisture proof pouches.

Thank you for this insightful comment. Initially we considered the freeze-drier itself as a potential roadblock for people attempting to lyophilize their open access reactions. However, our collaborators in Ghana have in fact a similar machine in their lab, which is a common laboratory freeze-dryer used in labs performing protein purification. While we agree that a more industrial machine would be of great utility in enabling higher throughput manufacturing, we had no problems in freeze-drying thousands of reactions in single-use tubes with this method, demonstrating the possibility to decentralize manufacturing of open source diagnostics.

The color transformations for the HNB color change are interesting, in that I hadn't seen that before. I assume this is purely to aid the human eye in visualizing the difference, vs the original RGB image, but to me it seems this is 'after-the-fact', i.e. there is no new information being generated by the image processing, but rather a magnification or rescaling and false-coloring of the difference in hue, but this is all accomplished by image processing or machine vision that is independent of human perception.

To enhance the sometimes hard-to-appreciate color difference in positive vs negative HNB-LAMP reactions, i.e. due to different lighting conditions or for colorblind people, we previously developed the simple color conversion tool (colorimetry.net) to more easily distinguish between positives and negatives (first published by Kellner et al., 2022). The reviewer is correct in stating that it is not generating new information, and it is reliant on the presence of the color change in the first place (e.g. a robust LAMP reaction happened) and on a good image (quality camera and lighting). We found the colorimetry.net tool to

effectively combat the visually relatively weak HNB color change. We hope that this encourages more researchers to use HNB, especially for raw input samples where phenol red is not advised because of sample pH.

Reviewer #2 (Comments to the Authors (Required)):

This paper by the Brennecke and Pauli labs describes the development and validation of a sensitive, robust RT-LAMP-based assay for the detection of SARS-CoV-2 RNA with good to excellent sensitivity. Unlike the plethora of RT-LAMP papers that have emerged in the aftermath of the pandemic, this paper merits publication for several compelling reasons:

- The paper addresses the challenge of creating a field-compatible RT-LAMP assay using only non-proprietary, low-cost ingredients.
- It provides a sample lysis/processing solution for direct testing that appears to be equivalent to commercial (proprietary) formulations.
- It describes procedures for direct testing of swab and gargle samples with good sensitivity.
- It details an enzyme mixture for RT-LAMP that can be self-produced and demonstrates sensitivity comparable to commercial formulations.
- It incorporates measures to protect against contamination, a critical consideration in non-laboratory settings.
- The procedures developed include lyophilized formulations for scenarios where a continuous cold chain for enzyme/reagent delivery cannot be guaranteed.
- The work is executed to a high standard.
- The work emphasizes reproducibility and applicability, including a webpage with detailed supplementary information.
- The study constitutes a valuable resource for future pandemics.

We thank the reviewer for their positive assessment of our work.

The 'downside' of the method is that it has not yet been applied to a large set of samples in a SARS-CoV-2 pandemic setting (and it is likely never will be - which, in fact, is good news!). However, new pandemics may arise, and this work will undoubtedly prove very useful in such scenarios in the future.

We agree with the reviewer that experiments presented here do not represent a large study focused on clinical performance. While such a study would indeed be highly beneficial to the field, it unfortunately goes beyond the scope of our manuscript in which we aimed to develop a simple and robust assay as well as general workflows for future open-source point-of-care

diagnostics in low resource settings where antigen testing is unavailable and molecular diagnostics is unreachable.

What I find missing is one or two sections discussing critical points to prepare future users (which may be molecular biologists with little/zero experience in diagnostics) who might need to rapidly adapt the method to new scenarios:

We thank the reviewer for their constructive comment. We have in fact built a webpage rtlamp.org where we provide protocols and practical tips, and discuss implementation for people with little or no molecular biology expertise. We highlight this website more clearly in our revised version of the manuscript.

- Primer design is crucial-good LAMP primers make all the difference (and designing them is notoriously challenging).

We fully agree that LAMP primer design is one of the most important aspects of a successful LAMP testing strategy. We did not develop any new primers in the course of this project but instead tested and used a large array of published primers and have observed remarkable differences in performance of different primer pairs. Since our study did not aim to address primer design itself as there are already numerous papers on this topic, we have decided to not address this in the discussion as it does not relate to the main points of the paper.

- The fact that a sensitivity of CT=31 may suffice for one application but not for others, such as acute testing of symptomatic patients versus surveillance testing in a low-incidence setting.

We agree that the required sensitivity of a diagnostic assay depends on its intended use—whether for acute diagnosis, surveillance, or pooled testing. While RT-LAMP may be less sensitive than RT-qPCR, its rapid turnaround enables timely identification of infectious individuals. In outbreak scenarios, the speed of detection can be as critical as sensitivity: The frequency and immediacy of RT-LAMP testing can help compensate for its comparatively lower sensitivity. Alternatively, we have developed bead-LAMP, which combines nucleic acid capture with direct RT-LAMP assays, effectively increasing the amount of input material. This results in an approximately 10- to 50-fold increase in assay sensitivity, making it well-suited for applications requiring higher sensitivity or pooled testing. We have added a sentence discussing this, along with the point raised above.

- Different viruses/pathogens may have varying requirements for sensitivity, as viral titers could generally be lower.

We thank the reviewer for their comment. Viral titers for some diseases may be lower, likewise some pathogens might be more easily detectable by RT-LAMP or LAMP, either due to a difference in nucleic acid detected (e.g. RNA vs DNA) or primers being more efficient for some sequences.

- Including a short perspective on potential future improvements would further strengthen the paper.

We believe that the main future perspectives for open source RT-LAMP lie in real world applications. Field studies on a range of pathogens in different settings would make the clearest case for open diagnostics and also for the identification of potential limitations of this approach. We see opportunities for further improvements to the methods presented here in the optimization and quality control of enzyme production in low-tech settings. We acknowledge the importance of these points and have added a statement in our discussion about further possible improvements.

Altogether, the work can be published with minimal modifications.

We thank the reviewer for their constructive and helpful feedback!

Reviewer #3 (Comments to the Authors (Required)):

This study addresses significant challenges in achieving equitable access to molecular diagnostics. Firstly, the development of open-source, cost-effective assays for large-scale nucleic acid detection and secondly the creation of lyophilized reagent formulations for cold chain-independent storage and shipment.

The researchers have developed a sensitive, rapid and accurate RT-LAMP assay for virus detection (modelled on SARS-CoV-2), demonstrating non-proprietary enzymatic components that can be produced and lyophilized in standard laboratory settings - highly suitable for resource-limited environments. Data was clearly presented and strongly supportive.

The study contributes several advances to the field namely open-access enzymes and protocols, available on an open-access platform. These protocols empower laboratories to produce reagents independently, lowering reliance on commercial suppliers. The reagents and protocols were successfully tested on other respiratory viruses, highlighting their potential for pathogen surveillance.

Independent benchmarking studies in Vienna and Ghana confirmed the assay's high sensitivity and specificity using both contrived and clinical samples, demonstrating its robustness across varying conditions.

This work helps to pave the way for equitable access to diagnostics, strengthening global capabilities for pathogen surveillance and epidemic management.

We thank the reviewer for their positive assessment of our manuscript.

June 20, 2025

RE: Life Science Alliance Manuscript #LSA-2024-03167-TR

Dr. Andrea Pauli
Research Institute of Molecular Pathology
Vienna Biocenter (VBC)
Campus-Vienna-Biocenter 1
Vienna, Vienna 1030
Austria

Dear Dr. Pauli,

Thank you for submitting your revised manuscript entitled "A lyophilized open-source RT-LAMP assay for molecular diagnostics in resource-limited settings". As you will see below, Reviewer 1 is now satisfied. We would be happy to publish your paper in Life Science Alliance pending final revisions necessary to meet our formatting guidelines.

- Please be sure that the authorship listing and order is correct.
- Please add ORCID ID for secondary and third corresponding authors - they should have received instructions on how to do so.
- Please add the X and Bluesky handles of your host institute/organization, your own, and one of the authors in our system.
- Please create a "dummy account" for the Vienna Covid-19 Detection Initiative (VCDI) group with some email address and add it to the system as a coauthor.
- Please be sure that the authorship listing and order are correct.
- Please consult our manuscript preparation guidelines <https://www.life-science-alliance.org/manuscript-prep> and make sure your manuscript sections are in the correct order.
- Please use the [10 author names, et al.] format in your references (i.e., limit the author names to the first 10).
- Please add your main and supplementary figure legends to the main manuscript text after the references section.
- It is recommended to exclude figures from the manuscript text and upload them separately.
- Please add a callout for Figure S3C to your main manuscript text.

A. FINAL FILES:

B. MANUSCRIPT ORGANIZATION AND FORMATTING:

Sincerely,

Reviewer #1 (Comments to the Authors (Required)):

The authors have addressed my comments adequately and I believe the manuscript is suitable for publication in Life Science Alliance.

July 1, 2025

RE: Life Science Alliance Manuscript #LSA-2024-03167-TRR

Dr. Andrea Pauli
Research Institute of Molecular Pathology
Vienna Biocenter (VBC)
Campus-Vienna-Biocenter 1
Vienna, Vienna 1030
Austria

Dear Dr. Pauli,

Thank you for submitting your Methods entitled "A lyophilized open-source RT-LAMP assay for molecular diagnostics in resource-limited settings". It is a pleasure to let you know that your manuscript is now accepted for publication in Life Science Alliance. Congratulations on this interesting work.

DISTRIBUTION OF MATERIALS:

Again, congratulations on a very nice paper. I hope you found the review process to be constructive and are pleased with how the manuscript was handled editorially. We look forward to future exciting submissions from your lab.

Sincerely,
